# AREA: Attribute Extraction and Aggregation for CLIP-Based Class-Incremental Learning

Zhen-Hao Xie [* 1 2]   Yu-Cheng Shi [* 2]   Da-Wei Zhou [1 2]

## Abstract

Class-Incremental Learning (CIL) is important in building real-world learning systems. In CLIP-based CIL, the model performs classification by comparing similarity between visual and textual embeddings obtained from template prompts, *e.g.*, "a photo of a `[CLASS]`". This seemingly monolithic matching process can be decomposed into two conceptually distinct stages: *attribute extraction* and *attribute aggregation*. For example, a model may recognize `cat` using attributes such as *fur texture* and *whiskers*. When learning a new class like `car`, the model must extract additional attributes like *wheels* and adjust how they are aggregated in the shared representation space. However, since only data from the current task is available, incremental updates can bias both attribute extraction and aggregation toward new classes, leading to catastrophic forgetting. Therefore, we propose AREA for AttRibute Extraction and Aggregation in CLIP-based CIL. To stabilize extraction, we anchor class-level visual and textual attributes on the hyperspherical embedding space via principal geodesic analysis. To stabilize aggregation, we learn lightweight task-specific experts with scoring and residual refinement, regularized by a variational information bottleneck objective. During inference, we perform routing over task attribute manifolds via optimal transport for more concise prediction. Experiments show that AREA consistently outperforms SOTA methods. Code is available at https://github.com/LAMDA-CL/ICML2026-AREA.

---

[*]Equal contribution [1]School of Artificial Intelligence, Nanjing University, China [2]State Key Laboratory of Novel Software Technology, Nanjing University, China. Correspondence to: Da-Wei Zhou <zhoudw@lamda.nju.edu.cn>.

*Proceedings of the 43[rd] International Conference on Machine Learning*, Seoul, South Korea. PMLR 306, 2026. Copyright 2026 by the author(s).

## 1. Introduction

Class-Incremental Learning (CIL) (Zhou et al., 2024b; Kirkpatrick et al., 2017) aims to enable models to continuously acquire new knowledge over time while preserving performance on previously learned ones. In this scenario, training data arrive sequentially and past data are no longer accessible, making catastrophic forgetting (Serra et al., 2018; Ramasesh et al., 2021; Shi et al., 2021) a persistent challenge. Recently, increasing attention has shifted to leveraging pre-trained models (Zhou et al., 2024a; Dosovitskiy et al., 2021; Radford et al., 2021; Li et al., 2026a), as their strong generalization and semantic priors can substantially reduce the amount of task-specific adaptation needed and thereby improve stability under continual updates.

Building on this trend, vision–language models (Liu et al., 2023; Bai et al., 2023; Ning et al., 2025; Zhang et al., 2025) such as CLIP (Radford et al., 2021) have emerged as a powerful foundation for CIL (Zhou et al., 2025c;b). CLIP maps images and texts into a shared embedding space, making it well-suited for CIL due to its strong transferability. In vanilla CLIP-based CIL, classification is performed by directly matching an image embedding to class text embeddings induced by template prompts, *e.g.*, "a photo of a `[CLASS]`" via cosine similarity.

Nevertheless, treating CLIP classification as a single similarity match hides what the model actually relies on. In practice, a prediction is supported by multiple visual cues, and these cues are combined in the shared embedding space. For example, after learning `cat`, a model may base its predictions on visual attributes such as *fur texture*, *whiskers*, and *ear shape*. When a new class such as `car` arrives, the model must incorporate new attributes such as *wheels* and *windows* and recalibrate how all available attributes are aggregated to discriminate `car` from previously learned classes. Without access to old data, this adjustment biases toward the current task, overfitting the new attribute-combination pattern and implicitly diminishing the influence of old attributes.

In this work, we formalize this process by decomposing prediction into two functional components: **1)** attribute extraction, which extracts task-agnostic discriminative attributes from the input; and **2)** attribute aggregation, which

dynamically aggregates such attributes to produce final predictions. With this formulation, catastrophic forgetting can originate from two sources: the extracted attributes may drift across tasks, weakening the discriminative geometry of previously learned classes, while the aggregation may change how attributes are weighted, increasingly favoring newly introduced classes. Consequently, addressing catastrophic forgetting requires jointly stabilizing both attribute extraction and aggregation.

Hence, we propose AREA (AttRibute Extraction and Aggregation) to boost CLIP-based CIL regarding these steps. We first build class-level visual and textual anchors on the hypersphere using principal geodesic analysis, so that the extracted attribute structure remains fixed once a class is observed. On top of these anchors, AREA trains lightweight task experts that aggregate attribute evidence through scoring with residual refinement, and we regularize this aggregation with a variational information bottleneck objective to avoid shortcut-driven task-specific bias. At test time, we route queries to compatible experts via optimal transport on task attribute manifolds and combine their outputs in a soft mixture. Across multiple benchmarks, AREA achieves consistent state-of-the-art performance.

## 2. Related Work

**Class-Incremental Learning (CIL).** CIL (Zhou et al., 2024b; De Lange et al., 2021; Masana et al., 2022; Xie et al., 2026b; Lai et al., 2026) studies how to continually incorporate novel classes without access to previously seen data, while maintaining performance on all past classes. A broad spectrum of methods has been developed. Replay-based approaches (Rebuffi et al., 2017; Wu et al., 2019; Liu et al., 2020) mitigate forgetting by storing or generating a subset of past samples and replaying them during new-task training, often providing strong performance but can easily introduce privacy concerns. Regularization-based methods (Shi et al., 2022; Tan et al., 2024; Liu & Chang, 2025; Wen et al., 2025; Xie et al., 2026b;a) estimate parameter importance and penalize updates that would damage previously learned knowledge, thereby reducing drift at the cost of limited plasticity under large distribution shifts. Architectural isolation and expansion techniques (Wang et al., 2023; Yan et al., 2021; Douillard et al., 2022; Zheng et al., 2025) allocate dedicated parameters to new classes or tasks to reduce interference.

**CLIP-Based CIL.** CLIP-style vision-language models (Radford et al., 2021) offer a unified embedding space where images and texts are aligned via cosine similarity, enabling strong zero-shot transfer with minimal supervision. This property has made CLIP a natural backbone for CIL: many methods (Zhou et al., 2025c;b; Thengane et al., 2022) freeze the visual and textual encoders to preserve the

pre-trained alignment, and adapt to new classes by training only lightweight and task-specific components. Existing CLIP-based CIL approaches broadly fall into three families. Prompt-based methods (Singha et al., 2023; Lu et al., 2025) adapt the text side by selecting or generating prompts to better match new classes. Adapter-based methods (Zhou et al., 2025c; Huang et al., 2024; Zhou et al., 2025b; Wen et al., 2025; Cheng et al., 2026) inject small trainable modules into the encoders to steer representations while keeping most weights fixed. Low-rank adaptation approaches (Li et al., 2026b; Al Rahhal et al., 2025) update a compact low-rank subspace of parameters and leave the backbone intact, balancing plasticity and retention.

## 3. Preliminaries

**Class-Incremental Learning (CIL).** CIL studies the problem of learning a growing set of classes from a data stream while maintaining performance on all previously observed classes (Rebuffi et al., 2017; Wang et al., 2022b). We consider a sequence of training datasets $\{\mathcal{D}^1, \mathcal{D}^2, \ldots, \mathcal{D}^B\}$, where each task $\mathcal{D}^b = \{(\mathbf{x}_i, y_i)\}_{i=1}^{n_b}$ contains $n_b$ labeled samples. Each instance $\mathbf{x}_i \in \mathbb{R}^D$ belongs to a class $y_i \in \mathcal{Y}_b$, and the class sets are disjoint across tasks, *i.e.*, $\mathcal{Y}_b \cap \mathcal{Y}_{b'} = \varnothing$ for $b \neq b'$. We follow the *exemplar-free* CIL setting (Wang et al., 2022a), where no data from previous tasks can be stored or revisited. At task $b$, the learner only has access to $\mathcal{D}^b$ and must produce a unified classifier over all seen classes $\mathcal{Y}_{\leq b} = \bigcup_{k=1}^b \mathcal{Y}_k$. The objective is to learn a predictor $f : \mathcal{X} \to \mathcal{Y}_{\leq b}$ that minimizes the expected classification error over the joint distribution of all observed tasks:

$$f^* = \arg\min_{f \in \mathcal{H}} \mathbb{E}_{(\mathbf{x},y)\sim\mathcal{D}^1\cup\cdots\cup\mathcal{D}^b}\mathbb{I}\big(y \neq f(\mathbf{x})\big), \quad (1)$$

where $\mathcal{H}$ denotes the hypothesis space and $\mathbb{I}(\cdot)$ is the indicator function.

**CLIP-Based CIL.** In this work, we consider CIL built upon a pre-trained CLIP model (Radford et al., 2021), which consists of a visual encoder $g_v : \mathbb{R}^{D_v} \to \mathbb{R}^d$ and a textual encoder $g_t : \mathbb{R}^{D_t} \to \mathbb{R}^d$, mapping images and texts into a shared embedding space. Given an input image $\mathbf{x}$, its visual embedding is $g_v(\mathbf{x})$. Each class $c \in \mathcal{Y}_{\leq b}$ is associated with a textual description $\mathbf{w}_c$ of the form "a photo of a [CLASS]$_c$", where the embedding of this description is $g_t(\mathbf{w}_c)$ (Huang et al., 2024; Yu et al., 2024). CLIP performs classification by comparing visual and textual embeddings through cosine similarity:

$$p(y = c \mid \mathbf{x}) = \frac{\exp\big(\cos(g_v(\mathbf{x}), g_t(\mathbf{w}_c))/\tau\big)}{\sum_{c' \in \mathcal{Y}_{\leq b}} \exp\big(\cos(g_v(\mathbf{x}), g_t(\mathbf{w}_{c'}))/\tau\big)}, \quad (2)$$

where $\tau$ is a temperature parameter. The backbones are typically frozen, limiting adaptation to lightweight components.

In Eq. (2), classification is governed by the matching degree between visual and textual embeddings.

**Decomposed Prediction.** A key observation from Eq. (2) is that, while CLIP appears to classify by a single cosine matching, this direct similarity computation hides the fact that the decision is implicitly supported by multiple class-specific cues and by how these cues are weighted and composed in the shared embedding space. For any input $\mathbf{x}$ and class $c$, the cosine similarity $\cos(g_v(\mathbf{x}), g_t(\mathbf{w}_c))$ acts as a basic signal of how much $\mathbf{x}$ supports class $c$. More broadly, prediction can be decomposed into a two-step process: *extracting attributes* from the input and then *aggregating attributes* to form a final class representation.

To make the above decomposition explicit, we rewrite Eq. (2) in terms of attribute extraction and aggregation. For an input image $\mathbf{x}_i$, we assume it gives rise to a series of latent attributes $[\mathbf{u}_{i,1}, \mathbf{u}_{i,2}, \ldots, \mathbf{u}_{i,K}]$, where each $\mathbf{u}_{i,k} \in \mathbb{R}^d$ corresponds to a discriminative attribute extracted from the input image $\mathbf{x}_i$. Similarly, each class $c$ is associated with a series of class-level attributes $[\mathbf{v}_{c,1}, \mathbf{v}_{c,2}, \ldots, \mathbf{v}_{c,K}]$, where each $\mathbf{v}_{c,k} \in \mathbb{R}^d$ encodes how different attributes characterize class $c$. Prediction is performed by aggregating attribute-level similarities with learnable weights. Specifically, the probability $p(y = c \mid \mathbf{x}_i)$ for class $c$ can be defined as:

$$\frac{\exp\big(\cos(\sum_{k=1}^{K} \alpha_{i,k}\mathbf{u}_{i,k}, \sum_{k=1}^{K} \beta_{c,k}\mathbf{v}_{c,k})/\tau\big)}{\sum_{c' \in \mathcal{Y}_{\leq b}} \exp\big(\cos(\sum_{k=1}^{K} \alpha_{i,k}\mathbf{u}_{i,k}, \sum_{k=1}^{K} \beta_{c',k}\mathbf{v}_{c',k})/\tau\big)}, \quad (3)$$

where $\alpha_{i,k} \geq 0$ denotes the aggregation weight over the extracted $k$-th attribute of input $\mathbf{x}_i$, and $\beta_{c,k} \geq 0$ denotes the aggregation weight associated with class $c$, with $\sum_{k=1}^{K} \alpha_{i,k} = \sum_{k=1}^{K} \beta_{c,k} = 1$. Under this formulation, catastrophic forgetting can arise from two distinct sources. Extraction drift corresponds to changes in the extracted attribute $\mathbf{u}_{i,k}$ or class attribute $\mathbf{v}_{c,k}$ for previously learned classes. Aggregation drift corresponds to shifts in the aggregation weights $\alpha_{i,k}$ and $\beta_{c,k}$, which may increasingly emphasize attributes useful for newly introduced classes while suppressing those critical for old ones. Both types of drift may cause catastrophic forgetting.

## 4. Method

In this section, we present AREA to mitigate catastrophic forgetting by stabilizing attribute extraction and aggregation. We first anchor class-level visual and textual attributes on the hypersphere via PGA to prevent extraction drift. We then aggregate anchored attributes using scoring and residual refinement, regularized by a variational information bottleneck objective for stable and generalizable evidence. At inference time, we perform soft routing over task manifolds to select compatible task experts under cross-task overlap.

### 4.1. Extracting Multi-Modal Attributes

Following Eq. (3), we first stabilize attribute extraction by constructing geometry-aware class-level anchors that remain fixed once a class is observed. Therefore, we build a compact set of class-specific attributes and reuse them across subsequent tasks as stable references for later aggregation.

**Visual Attributes.** Conceptually, a class representation in hidden space can be decomposed from a single point to a small set of dominant directions around its prototype. We capture these directions as a $K$-dimensional attribute basis $V_c^{\text{vis}}$. In Euclidean space, one would obtain $V_c^{\text{vis}}$ by decomposing a covariance matrix, but on $\mathbb{S}^{d-1}$ the notion of variation must follow geodesics. This motivates principal geodesic analysis (PGA) (Fletcher et al., 2004), which computes a covariance in a locally linear neighborhood on the hypersphere and extracts its leading directions:

$$C_c^{\text{vis}} = U_c^{\text{vis}} \Sigma_c^{\text{vis}} U_c^{\text{vis}\top}, \quad V_c^{\text{vis}} = U_c^{\text{vis}}[:, :K]. \quad (4)$$

To construct $C_c^{\text{vis}}$ in a geometry-faithful way, when task $b$ arrives we collect normalized visual embeddings $\{\mathbf{x}_i\}_{i=1}^{n_c}$ for each class $c \in \mathcal{Y}_b$ using the frozen encoder $g_v$, where $\|\mathbf{x}_i\|_2 = 1$ implies $\mathbf{x}_i \in \mathbb{S}^{d-1}$. We first locate the class prototype on the hypersphere by the Fréchet mean:

$$\boldsymbol{\mu}_c^{\text{vis}} = \arg\min_{\mathbf{p} \in \mathbb{S}^{d-1}} \sum_{i=1}^{n_c} d^2(\mathbf{p}, \mathbf{x}_i; \mathbb{S}^{d-1}), \quad (5)$$

where $d(\mathbf{p}, \mathbf{x}; \mathbb{S}^{d-1}) = \arccos(\mathbf{p}^\top \mathbf{x})$ is the geodesic distance induced by cosine similarity. We then flatten the local neighborhood around $\boldsymbol{\mu}_c^{\text{vis}}$ by mapping each sample to the tangent space via the logarithmic map:

$$\mathbf{u}_i = \frac{\arccos(\boldsymbol{\mu}_c^{\text{vis}\top} \mathbf{x}_i)}{\sqrt{1 - (\boldsymbol{\mu}_c^{\text{vis}\top} \mathbf{x}_i)^2}} \big(\mathbf{x}_i - (\boldsymbol{\mu}_c^{\text{vis}\top} \mathbf{x}_i)\boldsymbol{\mu}_c^{\text{vis}}\big). \quad (6)$$

We then compute the tangent covariance $C_c^{\text{vis}} = \frac{1}{n_c} \sum_{i=1}^{n_c} \mathbf{u}_i \mathbf{u}_i^\top$, and finally obtain $V_c^{\text{vis}}$ via Eq. (4). The columns of $V_c^{\text{vis}}$ form an orthonormal basis in the tangent space and define a class-specific attribute subspace whose directions correspond to principal geodesics through $\boldsymbol{\mu}_c^{\text{vis}}$.

**Textual Attributes.** We anchor textual cues in the same manner. For each sample $\mathbf{x}_i$, we obtain a fine-grained caption $\mathbf{c}_i$ using an MLLM and fuse it with the class prompt $\mathbf{w}_c$ to form $h_t(\mathbf{x}_i) = \text{Norm}(g_t(\mathbf{w}_c) + g_t(\mathbf{c}_i))$, ensuring $h_t(\mathbf{x}_i) \in \mathbb{S}^{d-1}$. We then apply the same PGA procedure to the textual set $\mathcal{H}_c = \{h_t(\mathbf{x}_i)\}_{i=1}^{n_c}$ to obtain $(\boldsymbol{\mu}_c^{\text{txt}}, V_c^{\text{txt}})$.

**Discussions.** We instantiate the class attributes by the anchored bases $V_c^{\text{vis}}$ and $V_c^{\text{txt}}$ obtained in Eq. (4), where each column corresponds to one attribute direction (*i.e.*, $\mathbf{v}_{c,k}$ in Eq. (3)). By computing these bases with PGA on $\mathbb{S}^{d-1}$, the anchors respect the cosine-based hyperspherical structure of CLIP features and avoid geometry-induced distortion. Once

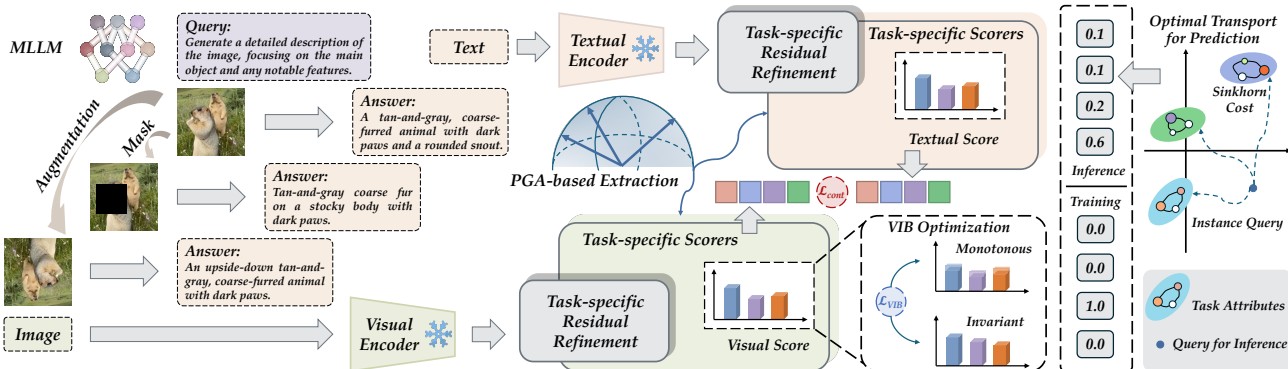

*Figure 1.* Overview of AREA. We freeze CLIP throughout training. For each incoming task, we build multi-modal class attributes on the hypersphere via PGA from visual features and caption-augmented text embeddings. Then we train a lightweight task expert that aggregates anchored attributes using scoring and residual refinement, regularized with a variational information bottleneck objective. At inference time, we use Sinkhorn optimal-transport routing over task attribute manifolds and softly combine the experts' predictions.

constructed, $V_c^{\text{vis}}$ and $V_c^{\text{txt}}$ are frozen and reused in later tasks, providing fixed attribute banks for aggregation and thereby mitigating extraction drift.

## 4.2. Aggregating Attributes

Following the acquisition of visual and textual attributes, *how could we effectively aggregate the attributes without drift?* To address this challenge, we introduce a dual-branch architecture that consists of two modules: the aggregation scoring and the residual refinement module, which are appended to the visual and textual encoders, respectively.

The aggregation scoring module aims to transform raw multi-modal attributes into structured and attribute-aware embeddings. For task $b$, we parameterize it with a score mapping branch $\mathcal{S}_b^{\text{vis}}, \mathcal{S}_b^{\text{txt}} \in \mathbb{R}^{d \times K}$ that projects visual and textual embeddings into score space, and a residual refinement branch $\mathcal{R}_b^{\text{vis}}, \mathcal{R}_b^{\text{txt}} \in \mathbb{R}^{d \times d}$ that provides complementary corrections. The resulting visual score embedding for sample $\mathbf{x}_i$ is computed as follows:

$$\mathcal{E}_v(\mathbf{x}_i) = V_c^{\text{vis}} \mathcal{S}_b^{\text{vis}}(g_v(\mathbf{x}_i)) + \mathcal{R}_b^{\text{vis}}(g_v(\mathbf{x}_i)), \quad (7)$$

where $\mathcal{S}_b^{\text{vis}}(g_v(\mathbf{x}_i))$ computes the visual score from the raw visual embedding, and $\mathcal{R}_b^{\text{vis}}(g_v(\mathbf{x}_i))$ provides the corresponding residual refinement to further improve this embedding. $V_c^{\text{vis}}$ are the fixed visual attributes of class $c$. Similarly, the textual score embedding is

$$\mathcal{E}_t(\mathbf{x}_i) = T_c^{\text{txt}} \mathcal{S}_b^{\text{txt}}(h_t(\mathbf{x}_i)) + \mathcal{R}_b^{\text{txt}}(h_t(\mathbf{x}_i)). \quad (8)$$

**Discussions.** Recall that Eq. (3) decomposes prediction into attribute extraction and aggregation, where $\alpha_{i,k}$ weights how much the $k$-th extracted attribute contributes for input $\mathbf{x}_i$. In our design, the aggregation scoring branch plays exactly this role: it produces a $K$-dimensional score vector that serves as a sample-specific weighting over the anchored attribute basis. Concretely, we interpret $\boldsymbol{\alpha}_i^{\text{vis}} = \mathcal{S}_b^{\text{vis}}(g_v(\mathbf{x}_i))$ and

$\boldsymbol{\beta}_i^{\text{txt}} = \mathcal{S}_b^{\text{txt}}(h_t(\mathbf{x}_i))$ as the aggregation weights, which are then used to combine the fixed attribute bases into an attribute-aware embedding.

## 4.3. Stabilized and Generalized Aggregation

The scoring and residual modules in Sec. 4.2 are designed to aggregate and refine attribute-aligned evidence for each incoming task. However, a key challenge in CIL is that the scorer is optimized on a task-specific distribution. Without proper regularization, it may exploit task-dependent shortcuts (Bassi et al., 2024) that are predictive only within the current task. This makes aggregation sensitive to input perturbations and gradually shifts the attribute evidence toward task-specific nuisances, inducing aggregation drift that interferes with previously learned classes.

To make attribute aggregation both stable across tasks and generalizable to unseen data, we formulate the learning objective using the variational information bottleneck principle. Our goal is to learn an aggregated attribute representation $\mathcal{Z}$ that is maximally predictive of the label $Y$ while being minimally redundant with the input $X$:

$$\min_\theta \mathcal{L}_{\text{VIB}} = \underbrace{-I(\mathcal{Z}; Y)}_{\text{Predictive}} + \beta \underbrace{I(\mathcal{Z}; X)}_{\text{Compressive}}, \quad (9)$$

where $I(\cdot; \cdot)$ denotes mutual information. In practice, we instantiate $\mathcal{Z}$ as the attribute-evidence scores produced by the aggregation module. However, directly optimizing Eq. (9) is intractable because both mutual-information terms depend on unknown data distributions and require high-dimensional integration. Instead, we optimize two tractable surrogates that reflect the two desiderata in Eq. (9): (i) valid prediction that discourages shortcut evidence, and (ii) invariant compression that suppresses view-sensitive noise.

Maximizing $I(\mathcal{Z}; Y)$ requires $\mathcal{Z}$ to encode reliable, label-relevant evidence. However, in continual learning, the scorer

may produce artificially strong evidence by exploiting task-specific artifacts. To discourage such spurious evidence, we introduce an intervention-based monotonicity constraint: when informative regions are partially corrupted, the attribute evidence should not spuriously increase.

Concretely, we construct an intervened view $\tilde{\mathbf{x}}_i$ by randomly occluding a contiguous region of $\mathbf{x}_i$. We sample an occlusion ratio $\rho \sim \mathrm{Unif}(\rho_{\min}, \rho_{\max})$ and an aspect ratio $\eta$ to determine the block size, then generate a binary mask $\mathbf{M}_i$. The intervened image is formed as:

$$\tilde{\mathbf{x}}_i = \mathbf{M}_i \odot \mathbf{x}_i + (1 - \mathbf{M}_i) \odot \boldsymbol{\epsilon}_i, \tag{10}$$

where $\boldsymbol{\epsilon}_i$ is random noise.[1] Let the aggregated attribute evidence be:

$$s(\mathbf{x}) = \mathcal{S}_b^{\mathrm{vis}}\big(g_v(\mathbf{x})\big) + \mathcal{S}_b^{\mathrm{txt}}\big(h_t(\mathbf{x})\big). \tag{11}$$

We then enforce interventional monotonicity by penalizing evidence increases under occlusion:

$$\mathcal{L}_{\mathrm{int}} = \Big| \max\Big(0, s(\tilde{\mathbf{x}}_i) - s(\mathbf{x}_i)\Big) \Big|_1. \tag{12}$$

This one-sided constraint discourages spurious evidence that emerges from corrupted inputs, encouraging the scorer to rely on intrinsic and attribute-aligned cues.

On the other hand, minimizing $I(\mathcal{Z}; X)$ encourages $\mathcal{Z}$ to discard input-specific details that are irrelevant to class identity. However, such details often correspond to high-entropy nuisance factors that vary across tasks. We therefore enforce invariance of attribute evidence across different transformations, so that $\mathcal{Z}$ retains only persistent attributes.

For each sample $\mathbf{x}_i$, we generate $M$ augmented views $\{\mathbf{x}_i^{(m)}\}_{m=1}^M$ using standard transformations $\mathcal{T}$.[2] Let $s_{i,m}^{\mathrm{vis}}$ and $s_{i,m}^{\mathrm{txt}}$ denote the view-specific visual and textual scores, and let $\bar{s}_i^{\mathrm{vis}}$ and $\bar{s}_i^{\mathrm{txt}}$ be their mean across views. We minimize the deviation from the view centroid:

$$\mathcal{L}_{\mathrm{comp}} = \sum_{m=1}^M \Big( \big\| s_{i,m}^{\mathrm{vis}} - \bar{s}_i^{\mathrm{vis}} \big\|_1 + \big\| s_{i,m}^{\mathrm{txt}} - \bar{s}_i^{\mathrm{txt}} \big\|_1 \Big). \tag{13}$$

By minimizing $\mathcal{L}_{\mathrm{comp}}$, the aggregation becomes insensitive to view-dependent perturbations, effectively compressing $\mathcal{Z}$ to preserve only task-invariant attribute evidence.

**Overall Objective.** Combining the above regularizers, our stabilized aggregation objective is:

$$\mathcal{L}_{\mathrm{stab}} = \lambda_{\mathrm{int}} \mathcal{L}_{\mathrm{int}} + \lambda_{\mathrm{comp}} \mathcal{L}_{\mathrm{comp}} + \mathcal{L}_{\mathrm{cont}}, \tag{14}$$

---

[1] We use i.i.d. Gaussian noise; implementation details are provided in the Appendix.

[2] We use random horizontal flipping and color jittering. Additional augmentation settings are reported in the Appendix.

where $\lambda_{\mathrm{int}}$ and $\lambda_{\mathrm{comp}}$ control the regularization strength, and $\mathcal{L}_{\mathrm{cont}}$ is the standard contrastive loss. Together, $\mathcal{L}_{\mathrm{int}}$ discourages shortcut evidence under intervention, while $\mathcal{L}_{\mathrm{comp}}$ enforces view-invariant compression, improving stability and generalization throughout continual learning.

## 4.4. Inference with Optimal Transport

During inference, the model must route an input $\mathbf{x}$ to the most relevant task-specific scorer. Standard task selection often relies on point-to-point similarity, which can be brittle when incremental tasks form overlapping manifolds in the embedding space. Instead, we cast task routing as an Optimal Transport (OT) problem (Peyré & Cuturi, 2019) that matches the query to each task's attribute manifold using a distributional transport cost.

We represent the query embedding as a Dirac source measure $\mu_{\mathbf{x}} = \delta_{g_v(\mathbf{x})}$. For each task $b$, we model its anchored visual attribute manifold as an empirical target measure supported on the basis vectors $\mathbf{v}_j$: $\nu_b = \frac{1}{N_b \times K} \sum_{j=1}^{N_b \times K} \delta_{\mathbf{v}_j}$, where $N_b$ is the number of classes in task $b$ and each class contributes $K$ basis vectors. We compute the entropic OT cost with cosine distance:

$$\mathbf{C}_{1,j}^b = 1 - \mathrm{sim}\left(g_v(\mathbf{x}), \mathbf{v}_j\right), \tag{15}$$

and obtain the Sinkhorn distance by:

$$\mathcal{W}_\varepsilon(\mu_{\mathbf{x}}, \nu_b) = \min_{\boldsymbol{\pi} \in \Pi(\mu_{\mathbf{x}}, \nu_b)} \langle \boldsymbol{\pi}, \mathbf{C}^b \rangle - \varepsilon H(\boldsymbol{\pi}), \tag{16}$$

where $\Pi(\mu_{\mathbf{x}}, \nu_b)$ denotes admissible transport plans and $H(\boldsymbol{\pi})$ is the entropy. The resulting $\mathcal{W}_\varepsilon$ provides a distribution-aware similarity between the query and task $b$. We then convert transport costs into routing probabilities via a Boltzmann distribution:

$$p(b \mid \mathbf{x}) = \frac{\exp\left(-\mathcal{W}_\varepsilon(\mu_{\mathbf{x}}, \nu_b)/\tau\right)}{\sum_{b'=1}^B \exp\left(-\mathcal{W}_\varepsilon(\mu_{\mathbf{x}}, \nu_{b'})/\tau\right)}, \tag{17}$$

and aggregate predictions in a mixture-of-experts manner:

$$\hat{y} = \arg \max_{c \in \mathcal{Y}_{\leq B}} \sum_{b=1}^B p(b \mid \mathbf{x}) \cdot \mathrm{sim}\left(\mathcal{E}_v^b(\mathbf{x}), \mathcal{E}_t^b(\mathbf{x}_c)\right), \tag{18}$$

where $\mathcal{Y}_{\leq B}$ is the union of class labels from all seen tasks. Through this way, we perform distribution-aware task selection on attribute manifolds, yielding robust soft routing and reducing misrouting under overlapping task representations.

## 5. Theoretical Analysis

This section shows that the Sinkhorn routing score is *Lipschitz continuous* with respect to input perturbations, ensuring that the induced task selection error is linearly bounded by the magnitude of *extraction drift*.

*Table 1.* Average and last performance comparison on nine datasets with **CLIP ViT-B/16** as the backbone. We report the results of all compared methods reproduced by their source code. The best and second-best results are highlighted in **bold** and underline, respectively.

| Methods | Aircraft | | | | Cars | | | | CIFAR | | | |
| --- | --- | --- | --- | --- | --- | --- | --- | --- | --- | --- | --- | --- |
| | B0 Inc10 | | B50 Inc10 | | B0 Inc10 | | B50 Inc10 | | B0 Inc10 | | B50 Inc10 | |
| | $\bar{\mathcal{A}}$ | $\mathcal{A}_B$ | $\bar{\mathcal{A}}$ | $\mathcal{A}_B$ | $\bar{\mathcal{A}}$ | $\mathcal{A}_B$ | $\bar{\mathcal{A}}$ | $\mathcal{A}_B$ | $\bar{\mathcal{A}}$ | $\mathcal{A}_B$ | $\bar{\mathcal{A}}$ | $\mathcal{A}_B$ |
| ZS-CLIP (Radford et al., 2021) | 26.66 | 17.22 | 21.70 | 17.22 | 82.60 | 76.37 | 78.32 | 76.37 | 81.81 | 71.38 | 76.49 | 71.38 |
| L2P (Wang et al., 2022b) | 47.19 | 28.29 | 44.07 | 32.13 | 76.63 | 61.82 | 76.37 | 65.64 | 82.74 | 73.03 | 81.14 | 73.61 |
| DualPrompt (Wang et al., 2022a) | 44.30 | 25.83 | 46.07 | 33.57 | 76.26 | 62.94 | 76.88 | 67.55 | 81.63 | 72.44 | 80.12 | 72.57 |
| CODA-Prompt (Smith et al., 2023) | 45.98 | 27.69 | 45.14 | 32.28 | 80.21 | 66.47 | 75.06 | 64.19 | 82.43 | 73.43 | 78.69 | 71.58 |
| RAPF (Huang et al., 2024) | 50.38 | 23.61 | 40.47 | 25.44 | 82.79 | 71.22 | 77.21 | 69.97 | 86.14 | 78.04 | 82.17 | 77.93 |
| MG-CLIP (Huang et al., 2025) | 48.33 | 32.34 | 26.28 | 13.02 | 88.21 | 79.73 | 84.58 | 79.62 | 89.74 | 82.78 | 85.62 | 81.26 |
| AREA | 71.03 | 61.78 | 66.64 | 63.10 | 97.77 | 96.17 | 97.06 | 96.05 | 89.24 | 83.69 | 85.98 | 83.42 |

| Methods | Food | | | | UCF | | | | CUB | | | |
| --- | --- | --- | --- | --- | --- | --- | --- | --- | --- | --- | --- | --- |
| | B0 In10 | | B50 Inc10 | | B0 In10 | | B50 Inc10 | | B0 Inc20 | | B100 Inc20 | |
| | $\bar{\mathcal{A}}$ | $\mathcal{A}_B$ | $\bar{\mathcal{A}}$ | $\mathcal{A}_B$ | $\bar{\mathcal{A}}$ | $\mathcal{A}_B$ | $\bar{\mathcal{A}}$ | $\mathcal{A}_B$ | $\bar{\mathcal{A}}$ | $\mathcal{A}_B$ | $\bar{\mathcal{A}}$ | $\mathcal{A}_B$ |
| ZS-CLIP (Radford et al., 2021) | 87.86 | 81.92 | 84.75 | 81.92 | 75.50 | 67.64 | 71.44 | 67.64 | 74.38 | 63.06 | 67.96 | 63.06 |
| L2P (Wang et al., 2022b) | 85.66 | 77.33 | 80.42 | 73.13 | 86.34 | 76.43 | 83.95 | 76.62 | 70.87 | 57.93 | 75.64 | 66.12 |
| DualPrompt (Wang et al., 2022a) | 84.92 | 77.29 | 80.00 | 72.75 | 85.21 | 75.82 | 84.31 | 76.35 | 69.89 | 57.46 | 74.40 | 64.84 |
| CODA-Prompt (Smith et al., 2023) | 86.18 | 78.78 | 80.98 | 74.13 | 87.76 | 80.14 | 83.04 | 75.03 | 73.12 | 62.98 | 73.95 | 62.21 |
| RAPF (Huang et al., 2024) | 88.57 | 81.15 | 85.53 | 81.17 | 92.28 | 80.33 | 90.31 | 81.55 | 79.09 | 62.77 | 72.82 | 62.93 |
| MG-CLIP (Huang et al., 2025) | 88.59 | 82.35 | 28.86 | 12.51 | 87.74 | 80.83 | 75.45 | 59.42 | 74.20 | 64.25 | 53.47 | 34.78 |
| AREA | 93.05 | 89.25 | 91.51 | 89.62 | 95.54 | 88.71 | 95.26 | 88.48 | 87.69 | 82.14 | 85.11 | 82.33 |

| Methods | ImageNet-R | | | | ObjectNet | | | | SUN | | | |
| --- | --- | --- | --- | --- | --- | --- | --- | --- | --- | --- | --- | --- |
| | B0 In20 | | B100 Inc20 | | B0 Inc20 | | B100 Inc20 | | B0 Inc30 | | B150 Inc30 | |
| | $\bar{\mathcal{A}}$ | $\mathcal{A}_B$ | $\bar{\mathcal{A}}$ | $\mathcal{A}_B$ | $\bar{\mathcal{A}}$ | $\mathcal{A}_B$ | $\bar{\mathcal{A}}$ | $\mathcal{A}_B$ | $\bar{\mathcal{A}}$ | $\mathcal{A}_B$ | $\bar{\mathcal{A}}$ | $\mathcal{A}_B$ |
| ZS-CLIP (Radford et al., 2021) | 83.37 | 77.17 | 79.57 | 77.17 | 38.43 | 26.43 | 31.12 | 26.43 | 79.42 | 72.11 | 74.95 | 72.11 |
| L2P (Wang et al., 2022b) | 75.97 | 66.52 | 72.82 | 66.77 | 51.40 | 39.39 | 48.91 | 42.83 | 82.82 | 74.54 | 79.57 | 73.10 |
| DualPrompt (Wang et al., 2022a) | 76.21 | 66.65 | 73.22 | 67.58 | 52.62 | 40.72 | 49.08 | 42.92 | 82.46 | 74.40 | 79.37 | 73.02 |
| CODA-Prompt (Smith et al., 2023) | 77.69 | 68.95 | 73.71 | 68.05 | 46.49 | 34.13 | 40.57 | 34.13 | 83.34 | 75.71 | 80.38 | 74.17 |
| RAPF (Huang et al., 2024) | 83.56 | 76.63 | 79.61 | 75.92 | 53.78 | 34.97 | 45.37 | 35.74 | 85.23 | 78.21 | 81.91 | 78.62 |
| MG-CLIP (Huang et al., 2025) | 83.18 | 77.20 | 50.47 | 35.37 | 51.41 | 40.90 | 25.00 | 12.39 | 53.71 | 41.62 | 26.58 | 12.64 |
| AREA | 86.36 | 81.15 | 82.83 | 81.03 | 61.02 | 49.20 | 54.15 | 49.73 | 86.36 | 80.26 | 83.91 | 80.98 |

Let $\mu_{\mathbf{x}} = \delta_{g_v(\mathbf{x})}$ be the source measure of the input and $\nu_b$ be the target attribute measure for task $b$. The routing decision relies on the regularized transport cost $\mathcal{W}_\varepsilon(\mu_{\mathbf{x}}, \nu_b)$. In CIL, *extraction drift* acts as a perturbation vector $\mathbf{\Delta}$, shifting the input from $\mathbf{z}$ to $\mathbf{z}' = \mathbf{z} + \mathbf{\Delta}$. A robust inference mechanism must ensure that such drift does not cause catastrophic fluctuations in the routing score.

**Theorem 5.1** (Stability of Sinkhorn Routing). *Assume the cost function $c(\cdot, \cdot)$ is Lipschitz continuous on the bounded domain. Then, the Sinkhorn routing score $\mathcal{S}_b(\mathbf{x}) = -\mathcal{W}_\varepsilon(\mu_{\mathbf{x}}, \nu_b)$ is Lipschitz continuous with respect to the input embedding. That is, for any drift $\mathbf{\Delta}$:*

$$|\mathcal{S}_b(\mathbf{z}) - \mathcal{S}_b(\mathbf{z} + \mathbf{\Delta})| \leq L \cdot \|\mathbf{\Delta}\|_2, \quad (19)$$

*where $L$ is a constant depending on the manifold curvature and regularization $\varepsilon$.*

Theorem 5.1 justifies OT-based routing: unlike hard decision boundaries that can flip under infinitesimal noise, the OT score varies smoothly with the input. The selection error is strictly bounded by the physical magnitude of the drift.

**Connection to Drift.** Extraction drift is mathematically equivalent to the perturbation $\mathbf{\Delta}$. Theorem 5.1 guarantees that if we control extraction drift via anchored attributes,

the routing stability is theoretically assured. Meanwhile, *aggregation drift* affects the discriminability within the selected task. While OT secures the routing, our VIB-based objective minimizes the generalization error bound. Full proofs are deferred to Appendix A.

# 6. Experiments

In this section, we evaluate AREA on nine benchmark datasets and compare it with state-of-the-art methods. We report incremental learning curves to quantify forgetting over long class sequences, and perform ablations to disentangle the contributions of attribute extraction and aggregation, complemented by hyperparameter sensitivity analyses to assess robustness. We further visualize the learned embedding geometry and task-routing behavior to qualitatively illustrate how AREA stabilizes attribute extraction and aggregation under continual updates. Additional results and implementation details are deferred to the Appendix E.

## 6.1. Implementation Details

**Datasets.** We follow (Zhou et al., 2025c;b) to evaluate the performance of AREA on nine benchmark datasets in CLIP-based CIL, *i.e.*, CIFAR100 (Krizhevsky, 2009),

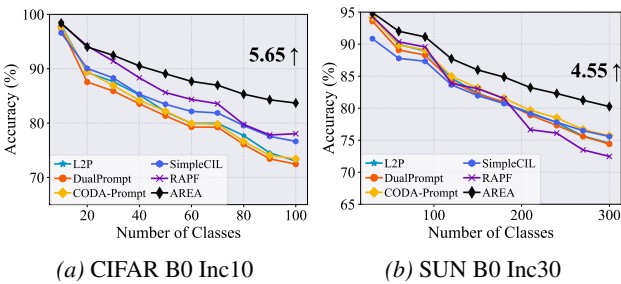

*(a)* CIFAR B0 Inc10     *(b)* SUN B0 Inc30

*Figure 2.* Performance curve of different methods under different settings. The relative improvement over the second-best method is annotated at the final incremental stage.

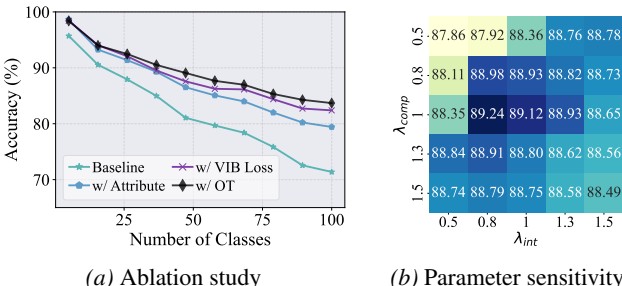

*(a)* Ablation study     *(b)* Parameter sensitivity

*Figure 3.* Ablation study and parameter sensitivity analysis.

CUB200 (Wah et al., 2011), ObjectNet (Barbu et al., 2019), ImageNet-R (Hendrycks et al., 2021), FGVCAircraft (Maji et al., 2013), StanfordCars (Krause et al., 2013), Food101 (Bossard et al., 2014), SUN397 (Xiao et al., 2010) and UCF101 (Soomro et al., 2012).

**Dataset Split.** Following (Rebuffi et al., 2017; Zhou et al., 2022), we use 'B-$m$ Inc-$n$' to split the classes in CIL. $m$ indicates the number of classes in the first stage, and $n$ represents that of every following stage. The dataset split is adapted following Zhou et al. (2025c;b). We follow Rebuffi et al. (2017) to randomly shuffle the class order with random seed 1993 for all experiments.

**Comparison Methods.** We first compare to a variety of recent SOTA pre-trained model-based CIL algorithms, *e.g.*, L2P (Wang et al., 2022b), DualPrompt (Wang et al., 2022a), CODA-Prompt (Smith et al., 2023), RAPF (Huang et al., 2024), and MG-CLIP (Huang et al., 2025). All methods are deployed with the same CLIP as initialization.

**Training Details.** All experiments are conducted on NVIDIA RTX 4090 GPUs using PyTorch (Paszke et al., 2019). We use C3Box (Sun & Zhou, 2026) to reproduce all compared methods. Following (Zhou et al., 2025c), we adopt CLIP with a ViT-B/16 backbone pre-trained on LAION-400M (Radford et al., 2021) across all methods to ensure a fair comparison. In AREA, we optimize the model with SGD using a batch size of 64 for 20 epochs. The learning rate is initialized to 0.05 and decayed according to the prescribed schedule. We set $\lambda_{mask} = 0.1$ and $\lambda_{cons} = 0.3$ for the masking and augmentation objectives, respectively. We employ GPT-5 (OpenAI, 2025) to generate captions describing fine-grained textual attributes.

### 6.2. Benchmark Comparison

We compare AREA against a broad set of state-of-the-art continual learning baselines on standard benchmarks, with quantitative results reported in Tab. 1 and performance curves illustrated in Fig. 2. AREA consistently achieves superior performance, outperforming most prior methods by more than 5% in both average accuracy and last-task

accuracy. Moreover, the performance curves indicate that our method exhibits a significantly smoother degradation trend compared to competing approaches. In particular, on the SUN Base0 Inc30 benchmark, AREA begins to surpass other methods by approximately 5% from Task 4 onward. This demonstrates that, as training progresses across tasks, the proposed Attribute Anchor effectively mitigates catastrophic forgetting caused by updates from new-task data.

In addition, our method shows clear advantages on more challenging datasets that are prone to severe inter-task confusion, such as ObjectNet and Aircraft. These results suggest that the introduced VIB Loss encourages the model to focus on key discriminative visual information, while the OT-based inference mechanism enables more accurate module selection, thereby further reducing task interference.

**Ablation Study.** We ablate each component on CIFAR100 B0 Inc10 setting in Fig. 3a. **Baseline** reports ZS-CLIP accuracy as classes accumulate and degrades sharply under the task-wise distribution shift. **w/ Attribute** adds our Attribute Anchors and multi-modal aggregation, yielding a large gain by providing stable class references and learnable aggregation. **w/ VIB Loss** further regularizes training with Eq. (9), encouraging label-relevant evidence and suppressing nuisance information, leading to additional improvement. Finally, **w/ OT** equips inference with OT-based routing, reducing cross-task interference under overlapping representations and achieving the best overall performance.

### 6.3. Further Analysis

**Hyperparameter Robustness.** We conduct a sensitivity study on CIFAR100 to evaluate the stability of AREA against key hyperparameters: the intervention loss weight $\lambda_{int}$ and the compression loss weight $\lambda_{comp}$. We sweep both weights across {0.5, 0.8, 1.0, 1.3, 1.5}. Fig. 3b reports the final average accuracy. We observe that performance remains consistently high across a broad range of $\lambda_{int}$ and $\lambda_{comp}$, peaking at $\lambda_{int} = 0.8, \lambda_{comp} = 1.0$. These results indicate that AREA is robust to hyperparameter change.

**Semantic Alignment of Attribute Anchoring.** To validate the effectiveness of using PGA for extracting attribute an-

*Table 2.* Average and last performance comparison on three datasets with annotations generated by different MLLMs. We adopt LLaVA-v1.6-34b as the additional MLLM. The best results are highlighted in **bold**.

| Methods | Aircraft B0 Inc10 | | | | CIFAR B0 Inc10 | | | | CUB B0 Inc20 | | | |
|---|---|---|---|---|---|---|---|---|---|---|---|---|
| | GPT5-Generated | | LLaVA-Generated | | GPT5-Generated | | LLaVA-Generated | | GPT5-Generated | | LLaVA-Generated | |
| | $\bar{\mathcal{A}}$ | $\mathcal{A}_B$ | $\bar{\mathcal{A}}$ | $\mathcal{A}_B$ | $\bar{\mathcal{A}}$ | $\mathcal{A}_B$ | $\bar{\mathcal{A}}$ | $\mathcal{A}_B$ | $\bar{\mathcal{A}}$ | $\mathcal{A}_B$ | $\bar{\mathcal{A}}$ | $\mathcal{A}_B$ |
| ZS-CLIP (Radford et al., 2021) | 26.66 | 17.22 | 26.13 | 17.06 | 81.81 | 71.38 | 82.05 | 71.49 | 74.38 | 63.06 | 73.05 | 63.15 |
| RAPF (Huang et al., 2024) | 50.38 | 23.61 | 49.86 | 23.20 | 86.14 | 78.04 | 85.97 | 78.84 | 79.09 | 62.77 | 78.97 | 61.84 |
| AREA | **71.03** | **61.78** | **70.89** | **60.95** | **89.24** | **83.69** | **88.98** | **83.24** | **87.69** | **82.14** | **86.86** | **81.22** |

*Table 3.* Top text tokens with the highest similarity to the anchors corresponding to the attributes, generated via PGA.

| # | PGA anchors (Similarity) |
|---|---|
| envelope | red (0.24), inscription (0.22) |
| hand mirror | pavement (0.21), tan (0.18) |
| jeans | blueberries (0.28), bluejays (0.23) |
| leaf | tiles (0.26), emerald (0.22) |
| banana | yellow (0.29), carrot (0.24) |
| box | sandstone (0.25), tile (0.20) |

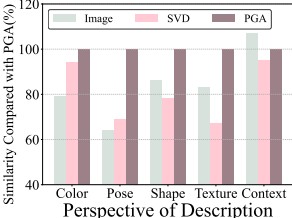

*(a)* Similarity comparison.

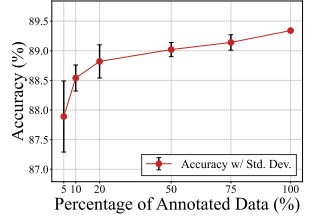

*(b)* Annotation proportion.

*Figure 4.* Analysis of PGA and annotation.

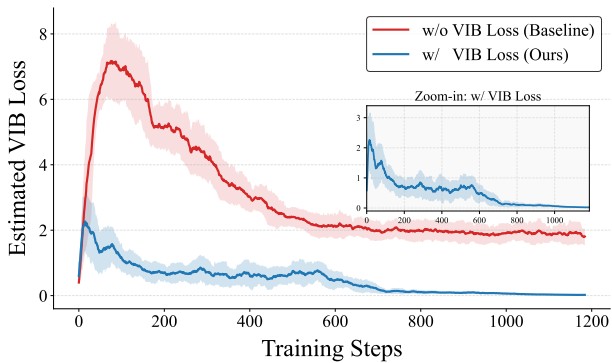

*Figure 5.* Estimated VIB Loss during the training on CIFAR100.

chors, we first employ a large language model to generate five fine-grained descriptions for each category in Object-Net, focusing on attributes such as color and shape. We then compute the similarity between the text embeddings of these fine-grained descriptions and (1) the visual features of the original images, (2) anchors extracted via SVD, and (3) anchors transformed by PGA. The similarities are averaged within each category, and the results are shown in Fig. 4a. As can be observed, compared to the anchors extracted by SVD, the PGA-based anchors are more effective at capturing intrinsic fine-grained attributes within each category. This capability enables the model to better preserve category-specific knowledge and effectively mitigates catastrophic forgetting, thereby validating the rationality of the proposed PGA-based multi-modal attribute extraction.

**Data Efficiency of MLLM Annotation.** Since acquiring MLLM-generated captions for every image incurs computational cost, we investigate the impact of partial textual supervision. We use 5 different seeds to randomly sample {5%, 10%, 20%, 50%, 75%, 100%} of the training data to generate captions and use the extracted anchors to train AREA on CIFAR-100. As shown in Fig. 4b, the performance gap between using 100% and 20% annotation is negligible, with a drop of less than 1%. Even with only 5% MLLM coverage, AREA can achieve a competitive accuracy of 87.9%. This suggests that the *Anchored Attribute Extraction* module

is highly data-efficient, capable of generalizing the learned textual subspace from a sparse set of enriched captions.

**Robustness to Annotation Sources.** Tab. 2 shows that AREA consistently achieves the best average and final accuracy across all three datasets under both GPT-5 and LLaVA-generated annotations (Liu et al., 2023). The gains are especially large on Aircraft and CUB, indicating stronger resistance to forgetting in harder settings. Performance remains nearly unchanged when switching the captioning MLLM, suggesting that AREA is robust to annotation sources.

**Semantic Interpretation of Learned Anchors.** Although we use the term "attributes" throughout this paper, we clarify that the learned components should not be interpreted as exact natural-language visual attributes of each class. Instead, they are better viewed as abstract feature directions in the representation space, which may exhibit only coarse semantic alignment with human-interpretable concepts. To examine this property, we probe the learned anchors by retrieving their nearest text tokens in the CLIP vocabulary. As shown in Tab. 3, the alignment with strict symbolic attributes is imperfect. For example, PGA retrieves loosely related concepts such as "red" and "inscription" for envelopes, reflecting an abstract color- or text-related direction rather than a concrete structural attribute of the class. These observations suggest that the PGA anchor subspace captures coarse semantic correlations at the representation level, rather than exact human-defined attributes.

**Variational Information Bottleneck Optimization.** Eq. (9) encourages task-relevant evidence while suppressing noise. On CIFAR100, we estimate the VIB loss with a sampling-based approximation during training, and report

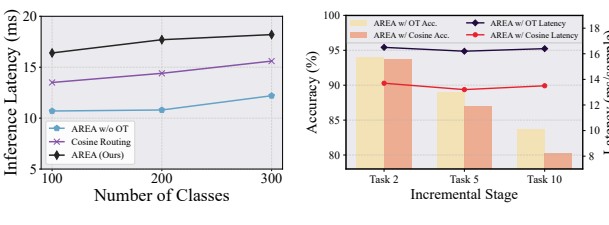

*(a)* Inference latency.  *(b)* Latency–accuracy trade-off.

*Figure 6.* Efficiency analysis of inference routing.

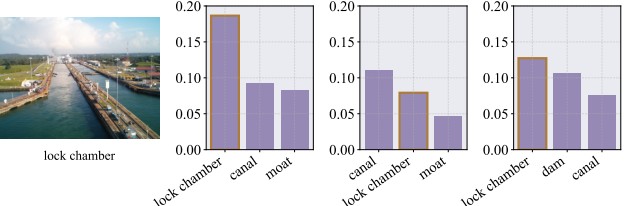

*Figure 7.* Prediction stability analysis for AREA. Bars denote the predicted class probabilities for the top-ranked classes.

the curves in Fig. 5. Compared to the baseline without the two indirect terms, AREA consistently achieves lower VIB loss, indicating that our objective better enforces the intended information bottleneck. We also observe a mild downward trend even without these terms, but the reduction is relatively smaller and less stable, motivating the VIB-based objective as an optimization target.

**Inference Efficiency and Routing Trade-off.** Since AREA performs OT-based routing over previously learned tasks and aggregates predictions from task-specific experts during inference, we further analyze its test-time efficiency and the necessity of OT routing. We report end-to-end latency and peak VRAM overhead on CIFAR-100, CUB-200, and SUN-300 on a single RTX 4090 with batch size 1. As shown in Fig. 6a, the inference latency of AREA increases mildly from 16.4 ms/sample to 18.2 ms/sample when the number of classes grows from 100 to 300. This is because AREA conducts routing over compact task-level attribute manifolds with batched Sinkhorn iterations, rather than exhaustive class-wise matching. These results suggest that the OT-based routing module introduces only a small constant overhead and does not become a practical bottleneck as the number of tasks or classes increases.

To further disentangle the contribution of distributional OT routing from the added inference complexity, we compare AREA with a simpler cosine-based routing baseline under the same CIFAR100 B0-Inc10 setting. As shown in Fig. 6b, cosine routing is slightly faster, but its accuracy drops more significantly in later incremental stages. At the final 100-class stage, OT routing only introduces an additional 2.9 ms/sample compared with cosine routing, while improving the final accuracy by 3.39%. This indicates a favorable accuracy–efficiency trade-off. The advantage comes from the fact that cosine routing performs point-to-point matching and is therefore more sensitive to local feature drift, whereas OT routing compares the query against the entire task-level attribute manifold. Such distribution-aware matching better captures task-level semantic structure and reduces cross-task misrouting in long-horizon incremental learning.

**Prediction Stability Analysis.** Fig. 7 illustrates the effect of Attribute Anchors on an SUN example. The first panel shows the prediction after training only on the first task,

where the model correctly predicts "lock chamber". The second panel reports the prediction after training on all tasks without anchors, where confidence shifts to competing classes and the correct class drops in rank. In contrast, the third panel shows that with anchors enabled, "lock chamber" remains top-ranked, indicating improved stability.

## 7. Conclusion

This paper tackles catastrophic forgetting in CLIP-based CIL by focusing on the underlying mechanism of attribute utilization. We stabilize the learning process by explicitly anchoring attribute extraction via a PGA-based mechanism and refining aggregation through a lightweight, learnable scoring module. By introducing interventional and compression terms to implicitly optimize the VIB loss, our approach effectively filters out redundancy, allowing the model to focus on the most discriminative attributes. Additionally, we consider an OT-based selection strategy to regulate attribute isolation and collaboration across tasks, further enhancing incremental stability. Extensive experiments across multiple benchmarks verify the superiority of AREA.

**Limitations.** Our study is limited to CLIP-based class-incremental learning with fixed pre-trained vision-language encoders. Its behavior under severe modality gaps, highly imbalanced data, or larger generative MLLMs remains underexplored and requires further study.

## Acknowledgments

This work was supported in part by the Basic Research Program of Jiangsu (BK20251251), NSFC (62506160), JSTJ-2025-147, Fundamental and Interdisciplinary Disciplines Breakthrough Plan of the Ministry of Education of China (No. JYB2025XDXM118), the 111 Center (No. B26023), and the Collaborative Innovation Center of Novel Software Technology and Industrialization.

## Impact Statement

This paper presents work whose goal is to advance the field of Machine Learning. There are many potential societal consequences of our work, none of which we feel must be specifically highlighted here.

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

## A. Proof of Stability (Theorem 5.1)

In this section, we provide the formal proof for Theorem 5.1, demonstrating the Lipschitz continuity of the Sinkhorn routing score with respect to the input visual embedding. This result theoretically guarantees that AREA is robust against *extraction drift* by bounding the routing error linearly with the magnitude of the drift.

### A.1. Preliminaries and Problem Setup

Recall the definition of the Sinkhorn distance (entropic optimal transport) between the source measure $\mu_{\mathbf{x}} = \delta_{\mathbf{z}}$ (where $\mathbf{z} = g_v(\mathbf{x}) \in \mathbb{S}^{d-1}$) and the target task measure $\nu_b = \sum_{j=1}^{N_b} w_j \delta_{\mathbf{v}_j}$ (where $\mathbf{v}_j \in \mathbb{S}^{d-1}$ are fixed attribute atoms). The regularized transport cost is defined as:

$$\mathcal{W}_\varepsilon(\mu_{\mathbf{x}}, \nu_b) = \min_{\boldsymbol{\pi} \in \Pi(\mu_{\mathbf{x}}, \nu_b)} \underbrace{\langle \boldsymbol{\pi}, \mathbf{C}(\mathbf{z}) \rangle - \varepsilon H(\boldsymbol{\pi})}_{\mathcal{L}(\boldsymbol{\pi}, \mathbf{z})}, \tag{20}$$

where $\mathbf{C}(\mathbf{z}) \in \mathbb{R}^{1 \times N_b}$ is the cost vector with elements $C_j(\mathbf{z}) = 1 - \mathbf{z}^\top \mathbf{v}_j$ (Cosine distance). The coupling space $\Pi(\mu_{\mathbf{x}}, \nu_b)$ is constrained by the marginals of $\mu_{\mathbf{x}}$ and $\nu_b$.

Our goal is to bound the variation of the routing score $\mathcal{S}_b(\mathbf{z}) = -\mathcal{W}_\varepsilon(\mu_{\mathbf{x}}, \nu_b)$ under a perturbation $\boldsymbol{\Delta}$ in $\mathbf{z}$. This is equivalent to bounding the norm of the gradient $\|\nabla_{\mathbf{z}} \mathcal{W}_\varepsilon\|_2$.

### A.2. Derivation of Lipschitz Continuity

We proceed by utilizing the envelope theorem for optimization problems to compute the sensitivity of the Sinkhorn distance.

**Lemma A.1** (Gradient via Danskin's Theorem). *Let $f(\mathbf{z}) = \min_{\boldsymbol{\pi} \in \Omega} \mathcal{L}(\boldsymbol{\pi}, \mathbf{z})$ be the optimal value function of the Sinkhorn problem. Since the entropic regularization term $-\varepsilon H(\boldsymbol{\pi})$ is strictly convex with respect to $\boldsymbol{\pi}$, there exists a unique optimal coupling $\boldsymbol{\pi}^*$. By **Danskin's Theorem**, the gradient of the value function with respect to the input parameter $\mathbf{z}$ is given by the gradient of the objective function evaluated at the unique minimizer:*

$$\nabla_{\mathbf{z}} \mathcal{W}_\varepsilon = \nabla_{\mathbf{z}} \mathcal{L}(\boldsymbol{\pi}^*, \mathbf{z}) = \sum_{j=1}^{N_b} \pi_j^* \nabla_{\mathbf{z}} C_j(\mathbf{z}). \tag{21}$$

*Proof.* The objective function $\mathcal{L}(\boldsymbol{\pi}, \mathbf{z})$ is continuous and differentiable with respect to $\mathbf{z}$. The feasible set $\Pi$ is compact. The strict convexity ensures the uniqueness of $\boldsymbol{\pi}^*$, satisfying the hypothesis for Danskin's Theorem, allowing the interchange of the differentiation and minimization operators. $\square$

**Lemma A.2** (Boundedness of Riemannian Gradient). *For any unit vectors $\mathbf{z}, \mathbf{v} \in \mathbb{S}^{d-1}$, let the cost function be the cosine distance $C(\mathbf{z}, \mathbf{v}) = 1 - \mathbf{z}^\top \mathbf{v}$. The magnitude of the gradient of $C$ with respect to $\mathbf{z}$ is bounded by 1.*

*Proof.* We consider the gradient in the ambient Euclidean space. The derivative is $\nabla_{\mathbf{z}}(1 - \mathbf{z}^\top \mathbf{v}) = -\mathbf{v}$. To be rigorous regarding the spherical constraint $\|\mathbf{z}\| = 1$, we project this gradient onto the tangent space $T_{\mathbf{z}} \mathbb{S}^{d-1}$ using the projection operator $\mathbf{P}_{\mathbf{z}} = \mathbf{I} - \mathbf{z}\mathbf{z}^\top$:

$$\nabla_{\mathbb{S}^{d-1}} C = \mathbf{P}_{\mathbf{z}}(-\mathbf{v}) = -\mathbf{v} + (\mathbf{z}^\top \mathbf{v})\mathbf{z}. \tag{22}$$

We compute the squared norm of the Riemannian gradient:

$$\|\nabla_{\mathbb{S}^{d-1}} C\|_2^2 = \| - \mathbf{v} + \cos\theta \cdot \mathbf{z}\|_2^2 \tag{23}$$

$$= \|\mathbf{v}\|_2^2 + \cos^2\theta \|\mathbf{z}\|_2^2 - 2\cos\theta(\mathbf{z}^\top \mathbf{v}) \tag{24}$$

$$= 1 + \cos^2\theta - 2\cos^2\theta \tag{25}$$

$$= 1 - \cos^2\theta = \sin^2\theta \leq 1. \tag{26}$$

Thus, the gradient norm is bounded by 1 everywhere on the manifold. $\square$

**Proof of Theorem 5.1.** Combining Lemma A.1 and Lemma A.2, we bound the gradient of the Sinkhorn distance:

$$\|\nabla_{\mathbf{z}} \mathcal{W}_\varepsilon\|_2 = \left\| \sum_{j=1}^{N_b} \pi_j^* \nabla_{\mathbf{z}} C_j(\mathbf{z}) \right\|_2 . \tag{27}$$

By applying the Triangle Inequality and substituting the bound from Lemma A.2:

$$\|\nabla_{\mathbf{z}} \mathcal{W}_\varepsilon\|_2 \le \sum_{j=1}^{N_b} |\pi_j^*| \cdot \|\nabla_{\mathbf{z}} C_j(\mathbf{z})\|_2 \le \sum_{j=1}^{N_b} \pi_j^* \cdot 1. \tag{28}$$

Since $\boldsymbol{\pi}^*$ is a valid transport plan, it must satisfy the marginal constraint for the source measure $\mu_{\mathbf{x}}$. Given that $\mu_{\mathbf{x}}$ is a Dirac measure with unit mass, the total mass transported is conserved: $\sum_{j=1}^{N_b} \pi_j^* = 1$. Therefore:

$$\|\nabla_{\mathbf{z}} \mathcal{W}_\varepsilon\|_2 \le 1. \tag{29}$$

Since the gradient norm is globally bounded by a constant $L = 1$, by the Mean Value Theorem on Riemannian manifolds, the function $\mathcal{W}_\varepsilon(\cdot, \nu_b)$ is 1-Lipschitz continuous. Consequently, for any extraction drift $\boldsymbol{\Delta}$ such that $\tilde{\mathbf{z}} = \mathbf{z} + \boldsymbol{\Delta}$:

$$|\mathcal{S}_b(\mathbf{z}) - \mathcal{S}_b(\tilde{\mathbf{z}})| = |\mathcal{W}_\varepsilon(\mu_{\mathbf{z}}, \nu_b) - \mathcal{W}_\varepsilon(\mu_{\tilde{\mathbf{z}}}, \nu_b)| \le 1 \cdot \|\boldsymbol{\Delta}\|_2. \tag{30}$$

This confirms that the routing error is linearly bounded by the drift magnitude, completing the proof. $\square$

# B. Detailed procedure of Sec. 4.1

This appendix provides the mathematical derivations and implementation specifics of the Principal Geodesic Analysis (PGA) introduced in Sec. 4.1. While standard Principal Component Analysis (PCA) assumes data resides in a Euclidean space, CLIP embeddings are normalized to the unit hypersphere $\mathbb{S}^{d-1}$. Applying Euclidean statistics directly ignores the manifold's curvature, leading to geometric distortion.

To address this, we employ PGA to generalize PCA to Riemannian manifolds. The procedure consists of three sequential steps: (1) estimating the intrinsic mean (Fréchet Mean), (2) mapping data to the tangent space via the Logarithmic Map, and (3) performing eigendecomposition in the tangent space.

### B.1. Step 1: Approximating the Fréchet Mean

The first step is to locate the center of the class distribution on the manifold. The Fréchet mean $\boldsymbol{\mu}_c$ is defined as the point minimizing the sum of squared geodesic distances:

$$\boldsymbol{\mu}_c^{\text{vis}} = \underset{\mathbf{p} \in \mathbb{S}^{d-1}}{\arg\min} \sum_{i=1}^{n_c} d^2(\mathbf{p}, \mathbf{x}_i; \mathbb{S}^{d-1}). \tag{31}$$

Unlike in Euclidean space, there is no closed-form solution for the global minimum on a generic sphere. However, since the visual features within a specific semantic class are highly concentrated (i.e., contained within a small geodesic ball), the arithmetic mean provides an extremely close approximation when projected back onto the sphere. In our implementation, we compute the initialized prototype as:

$$\boldsymbol{\mu}_c^{\text{vis}} \approx \frac{\sum_{i=1}^{n_c} \mathbf{x}_i}{\left\| \sum_{i=1}^{n_c} \mathbf{x}_i \right\|_2}. \tag{32}$$

### B.2. Step 2: Riemannian Logarithmic Map

This step corresponds to Eq. (6) in the main text and is crucial for flattening the curved geometry. We map the data samples $\mathbf{x}_i \in \mathbb{S}^{d-1}$ to the tangent space $T_{\boldsymbol{\mu}_c} \mathbb{S}^{d-1}$ at the mean. Geometrically, the tangent space is the hyperplane orthogonal to $\boldsymbol{\mu}_c$:

$$T_{\boldsymbol{\mu}_c} \mathbb{S}^{d-1} = \{\mathbf{v} \in \mathbb{R}^d \mid \mathbf{v}^\top \boldsymbol{\mu}_c = 0\}. \tag{33}$$

The Riemannian Logarithmic map, denoted as $\text{Log}_{\boldsymbol{\mu}_c}(\mathbf{x}_i)$, maps a point $\mathbf{x}_i$ on the sphere to a vector $\mathbf{u}_i$ in the tangent plane. The vector $\mathbf{u}_i$ preserves the distance from the mean, such that $\|\mathbf{u}_i\|_2 = d(\boldsymbol{\mu}_c, \mathbf{x}_i)$.

**Derivation.** Let $\theta_i = \arccos(\boldsymbol{\mu}_c^\top \mathbf{x}_i)$ be the geodesic distance (angle) between the mean and the sample. First, we compute the orthogonal projection of $\mathbf{x}_i$ onto the tangent plane using the Gram-Schmidt process:

$$\mathbf{x}_{i,\perp} = \mathbf{x}_i - (\boldsymbol{\mu}_c^\top \mathbf{x}_i)\boldsymbol{\mu}_c. \tag{34}$$

The Euclidean length of this projection is $\|\mathbf{x}_{i,\perp}\|_2 = \sin(\theta_i) = \sqrt{1 - (\boldsymbol{\mu}_c^\top \mathbf{x}_i)^2}$. To obtain the correct tangent vector $\mathbf{u}_i$, we must normalize $\mathbf{x}_{i,\perp}$ to unit length (giving the direction) and then scale it by the true geodesic distance $\theta_i$ (giving the magnitude):

$$\mathbf{u}_i = \underbrace{\theta_i}_{\text{Magnitude}} \cdot \underbrace{\frac{\mathbf{x}_{i,\perp}}{\|\mathbf{x}_{i,\perp}\|_2}}_{\text{Direction}}. \tag{35}$$

Substituting the terms back, we obtain the formulation used in the main paper:

$$\begin{aligned}
\mathbf{u}_i &= \frac{\arccos(\boldsymbol{\mu}_c^{\text{vis}\top}\mathbf{x}_i)}{\sin(\arccos(\boldsymbol{\mu}_c^{\text{vis}\top}\mathbf{x}_i))} \left(\mathbf{x}_i - (\boldsymbol{\mu}_c^{\text{vis}\top}\mathbf{x}_i)\boldsymbol{\mu}_c^{\text{vis}}\right) \\
&= \frac{\arccos(\boldsymbol{\mu}_c^{\text{vis}\top}\mathbf{x}_i)}{\sqrt{1 - (\boldsymbol{\mu}_c^{\text{vis}\top}\mathbf{x}_i)^2}} \left(\mathbf{x}_i - (\boldsymbol{\mu}_c^{\text{vis}\top}\mathbf{x}_i)\boldsymbol{\mu}_c^{\text{vis}}\right).
\end{aligned} \tag{36}$$

*Numerical Stability:* When $\mathbf{x}_i \to \boldsymbol{\mu}_c$, both the numerator and denominator approach zero. In practice, if $1 - (\boldsymbol{\mu}_c^\top \mathbf{x}_i)^2 < \epsilon$, we approximate the scaling factor $\frac{\theta}{\sin\theta} \to 1$ via Taylor expansion to avoid numerical instability.

### B.3. Step 3: Tangent Covariance and Attribute Basis

Once the dataset is mapped to the tangent vectors $\mathcal{U} = \{\mathbf{u}_1, \ldots, \mathbf{u}_{n_c}\}$, the problem reduces to standard Euclidean statistics in $\mathbb{R}^d$. The sample covariance matrix in the tangent space is:

$$C_c^{\text{vis}} = \frac{1}{n_c} \sum_{i=1}^{n_c} \mathbf{u}_i \mathbf{u}_i^\top. \tag{37}$$

We then perform Singular Value Decomposition (SVD) on $C_c^{\text{vis}}$ to obtain the eigenvectors $U_c^{\text{vis}}$ and eigenvalues $\Sigma_c^{\text{vis}}$. We select the top-$K$ eigenvectors $V_c^{\text{vis}} = U_c^{\text{vis}}[:,:K]$ to form the visual attribute basis. These vectors represent the principal geodesic directions that capture the maximum variance of the class distribution on the hypersphere. The same procedure is applied to the textual embeddings to obtain $V_c^{\text{txt}}$.

## C. Pseudocode

We summarize the training and inference procedures of AREA in Alg. 1 and Alg. 2. Training proceeds sequentially over tasks. For each new task, we first construct and freeze multi-modal class anchors via PGA (Sec. 4.1), then learn a lightweight task expert with stabilized aggregation (Sec. 4.2–Sec. 4.3). At inference time, we perform OT-based soft routing over task manifolds and aggregate predictions in a mixture-of-experts manner (Sec. 4.4).

## D. Implementation Details for Stabilized Aggregation

In this section, we provide the specific implementation details regarding the intervention-based monotonicity constraint and the view-invariant compression objective introduced in Sec. 4.3.

### D.1. Intervention and Occlusion Generation

To compute the intervention loss $\mathcal{L}_{\text{int}}$ (Eq. 12), we generate the intervened view $\tilde{\mathbf{x}}_i$ by corrupting a contiguous region of the input image with random noise. This process follows a randomized block-erasing strategy.

For an input image of spatial dimensions $H \times W$, the binary mask $\mathbf{M}_i$ and the noise tensor $\boldsymbol{\epsilon}_i$ are generated as follows:

**Geometric Parameters:** We sample the occlusion area ratio $\rho$ and the aspect ratio $\eta$ from uniform distributions:

$$\rho \sim \text{Unif}(\rho_{\min}, \rho_{\max}), \quad \eta \sim \text{Unif}(\eta_{\min}, \eta_{\max}). \tag{38}$$

---

**Algorithm 1** Training AREA on Task $b$

---

**Require:** Task dataset $\mathcal{D}_b = \{(\mathbf{x}_i, y_i)\}$, frozen encoders $g_v, g_t$, MLLM captioner, previous anchors $\mathcal{A}_{<b}$
**Require:** Hyperparameters $K, M, (\rho_{\min}, \rho_{\max}), \lambda_{\text{int}}, \lambda_{\text{comp}}$
**Ensure:** New anchors $\mathcal{A}_b$, trained task expert $\mathcal{E}_b = \{\mathcal{S}_b^{\text{vis}}, \mathcal{R}_b^{\text{vis}}, \mathcal{S}_b^{\text{txt}}, \mathcal{R}_b^{\text{txt}}\}$

1: **Anchor multi-modal attributes (Sec. 4.1).**
2: **for** each new class $c \in \mathcal{Y}_b$ **do**
3:     Extract normalized visual embeddings using $g_v$ for samples of class $c$
4:     Compute visual anchors $(\boldsymbol{\mu}_c^{\text{vis}}, V_c^{\text{vis}})$ via PGA (Eqs. (4)–(6))
5:     Generate MLLM captions and form textual embeddings $h_t(\cdot)$
6:     Compute textual anchors $(\boldsymbol{\mu}_c^{\text{txt}}, V_c^{\text{txt}})$ via the same PGA procedure
7: **end for**
8: Freeze and store anchors $\mathcal{A}_b = \{(\boldsymbol{\mu}_c^{\text{vis}}, V_c^{\text{vis}}, \boldsymbol{\mu}_c^{\text{txt}}, V_c^{\text{txt}})\}_{c \in \mathcal{Y}_b}$
9: **Train stabilized aggregation expert (Sec. 4.2–Sec. 4.3).**
10: Initialize task expert parameters $\mathcal{E}_b$
11: **repeat**
12:     Sample a mini-batch $\mathcal{B} \subset \mathcal{D}_b$
13:     **for** each $(\mathbf{x}_i, y_i) \in \mathcal{B}$ **do**
14:         Compute score-based and residual-refined embeddings using Eqs. (7)–(8)
15:         Compute intervention-based monotonicity loss $\mathcal{L}_{\text{int}}$ (Eqs. (10)–(12))
16:         Generate $M$ augmented views and compute compression loss $\mathcal{L}_{\text{comp}}$ (Eq. (13))
17:         Compute contrastive loss $\mathcal{L}_{\text{cont}}$ (Eq. (14))
18:     **end for**
19:     Form total objective $\mathcal{L}_{\text{stab}}$ by Eq. (14)
20:     Update task expert $\mathcal{E}_b$ by minimizing $\mathcal{L}_{\text{stab}}$
21: **until** convergence
22: **return** $\mathcal{A}_b$ and $\mathcal{E}_b$

---

In our experiments, we set $\rho_{\min} = 0.02$, $\rho_{\max} = 0.4$, $\eta_{\min} = 0.3$, and $\eta_{\max} = 3.3$. Based on these sampled values, the height $h_e$ and width $w_e$ of the occlusion block are calculated as $h_e = \sqrt{HW\rho\eta}$ and $w_e = \sqrt{HW\rho/\eta}$. The block is placed at a random coordinate $(x_e, y_e)$ within the image such that the entire block lies inside the image boundaries.

**Noise Distribution:** The noise tensor $\boldsymbol{\epsilon}_i$ is generated from a standard independent and identically distributed (i.i.d.) Gaussian distribution:

$$\boldsymbol{\epsilon}_i \sim \mathcal{N}(\mathbf{0}, \mathbf{I}), \tag{39}$$

where $\mathbf{I}$ denotes the identity matrix. The dimensions of $\boldsymbol{\epsilon}_i$ match those of the input image $\mathbf{x}_i$. During the mixing process in Eq. 10, regions defined by the mask $\mathbf{M}_i$ are replaced by this Gaussian noise, serving as uninformative perturbations to test the scorer's stability.

### D.2. Augmentations for Invariant Compression

To optimize the compression objective $\mathcal{L}_{\text{comp}}$ (Eq. 13), we generate $M = 3$ augmented views for each image. We employ a stochastic data augmentation pipeline $\mathcal{T}$ designed to alter nuisance factors (e.g., color, orientation, scale) while preserving semantic attributes.

The specific transformations and their corresponding hyperparameters are listed in Tab. 4. For each view, transformations are applied sequentially with probability $p$.

## E. Additional Experiment Results

### E.1. Full Benchmark Comparisons

In this section, we provide a more comprehensive evaluation of different methods across a wide range of datasets and incremental settings. Specifically, we present the detailed performance curves for Aircraft, Cars, and CIFAR in Figure 8, CUB, Food, and ImageNet-R in Figure 9, and ObjectNet, SUN, and UCF101 in Figure 10. These results complement the

---

**Algorithm 2** Inference of AREA with OT-based Soft Routing

---

**Require:** Test image $\mathbf{x}$, frozen encoders $g_v, g_t$, anchors $\{\mathcal{A}_b\}_{b=1}^B$, experts $\{\mathcal{E}_b\}_{b=1}^B$
**Require:** OT parameters $(\varepsilon, \tau)$
**Ensure:** Predicted label $\hat{y}$

1: **OT-based routing (Sec. 4.4).**
2: Compute query embedding $g_v(\mathbf{x})$ and form the source measure (Eq. (15))
3: **for** each task $b = 1, \ldots, B$ **do**
4:     Construct the task attribute measure from anchored bases (as in Sec. 4.4)
5:     Compute Sinkhorn distance $\mathcal{W}_\varepsilon(\mu_{\mathbf{x}}, \nu_b)$ (Eq. (16))
6: **end for**
7: Convert transport costs to routing weights $p(b \mid \mathbf{x})$ (Eq. (17))
8: **Mixture-of-experts prediction (Sec. 4.4).**
9: **for** each task $b = 1, \ldots, B$ **do**
10:     Compute task-specific embeddings using the corresponding expert $\mathcal{E}_b$ (Eqs. (7)–(8))
11:     Compute task-wise similarity scores for all candidate classes
12: **end for**
13: Aggregate task-wise predictions with routing weights and output $\hat{y}$ (Eq. (18))
14: **return** $\hat{y}$

---

*Table 4.* Augmentation hyper-parameters used for Invariant Compression. The transformations are applied sequentially to generate diverse views.

| Transformation | Prob. ($p$) | Parameters |
|---|---|---|
| Random Resized Crop | 1.0 | Scale $\in [0.2, 1.0]$, Ratio $\in [3/4, 4/3]$ |
| Random Horizontal Flip | 0.5 | – |
| Color Jitter | 0.5 | Brightness: 0.4, Contrast: 0.4, Saturation: 0.4, Hue: 0.1 |
| Random Grayscale | 0.2 | – |
| Gaussian Blur | 0.2 | Kernel size $\in [0.1, 2.0]$ |

quantitative summaries provided in the main paper. As observed, AREA consistently outperforms other methods on different datasets and data splits (e.g., Base-0 vs. Base-50/100) by a substantial margin. We attribute this sustained performance to the synergistic design of our framework: 1) **Geometry-Aware Anchoring**: By employing *Principal Geodesic Analysis*, we respect the intrinsic hyperspherical structure of CLIP embeddings, ensuring that class prototypes remain stable and drift-free; 2) **Robust Evidence Aggregation**: The *Variational Information Bottleneck* objective acts as a semantic filter, suppressing task-specific shortcuts and noise through interventional regularization; and 3) **Distributional Task Routing**: The *Optimal Transport* mechanism replaces brittle point-estimates with global distributional matching, allowing for accurate expert selection even when task boundaries are ambiguous or overlapping.

### E.2. Robustness to Different Random Seeds

In Class-Incremental Learning (CIL), stochastic factors—such as the ordering of incoming classes—can significantly impact model performance. To evaluate the stability of our method, we conducted experiments using five distinct random seeds in CIFAR-100 B0 Inc10. The results with the average accuracy and standard deviation, presented in Fig. 11a, demonstrate that AREA consistently delivers high performance across different random initializations with minimal standard deviation. This confirms the strong robustness of AREA against stochastic variations inherent in the incremental learning process.

### E.3. Runtime Consumption

In this section, we present a detailed running time comparison of our method, AREA, against DualPrompt and RAPF across the CIFAR, Aircraft, ImageNet-R, and ObjectNet datasets. All experiments were conducted on a single NVIDIA 4090 GPU. As illustrated in Fig. 11b, AREA exhibits a marginal increase in training duration compared to the baselines. This

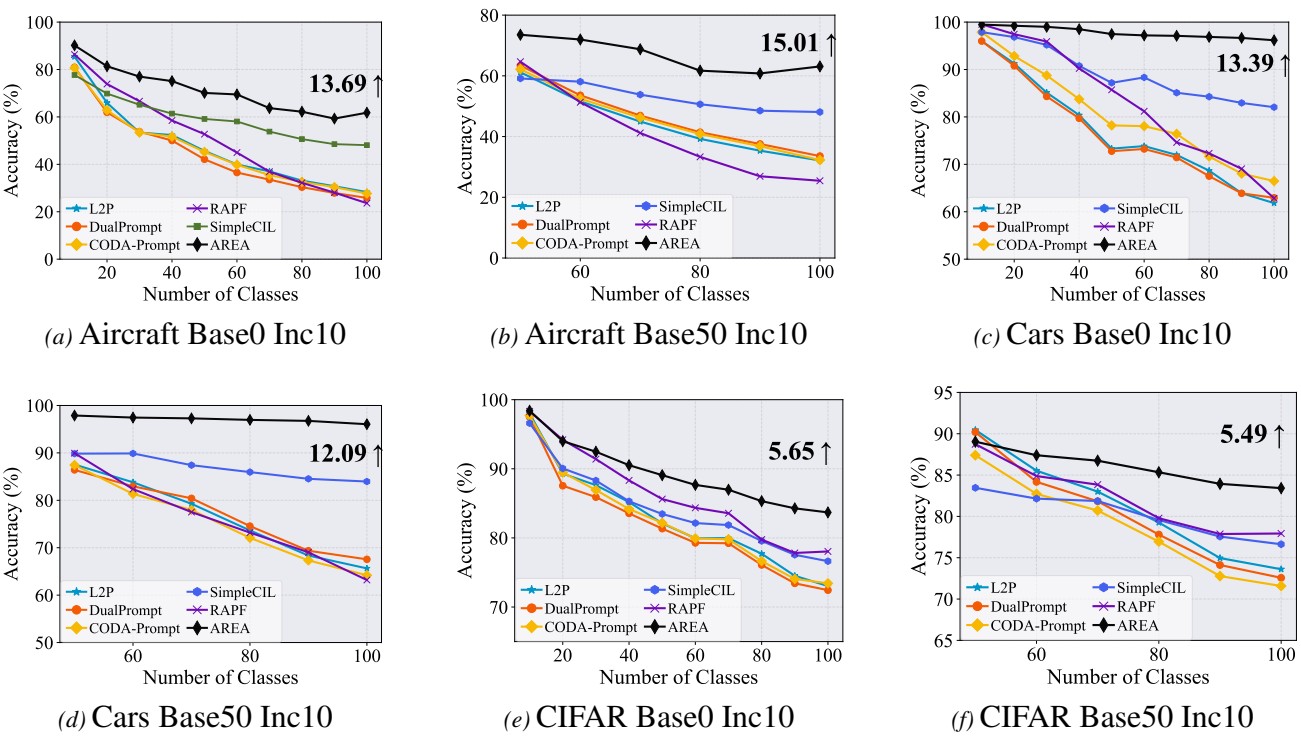

*(a)* Aircraft Base0 Inc10    *(b)* Aircraft Base50 Inc10    *(c)* Cars Base0 Inc10

*(d)* Cars Base50 Inc10    *(e)* CIFAR Base0 Inc10    *(f)* CIFAR Base50 Inc10

*Figure 8.* Incremental performance of different methods. We report the performance gap on Aircraft, Cars, and CIFAR datasets (Base0 and Base50 settings) after the last incremental stage of AREA and the runner-up method. All methods utilize the same pre-trained weights.

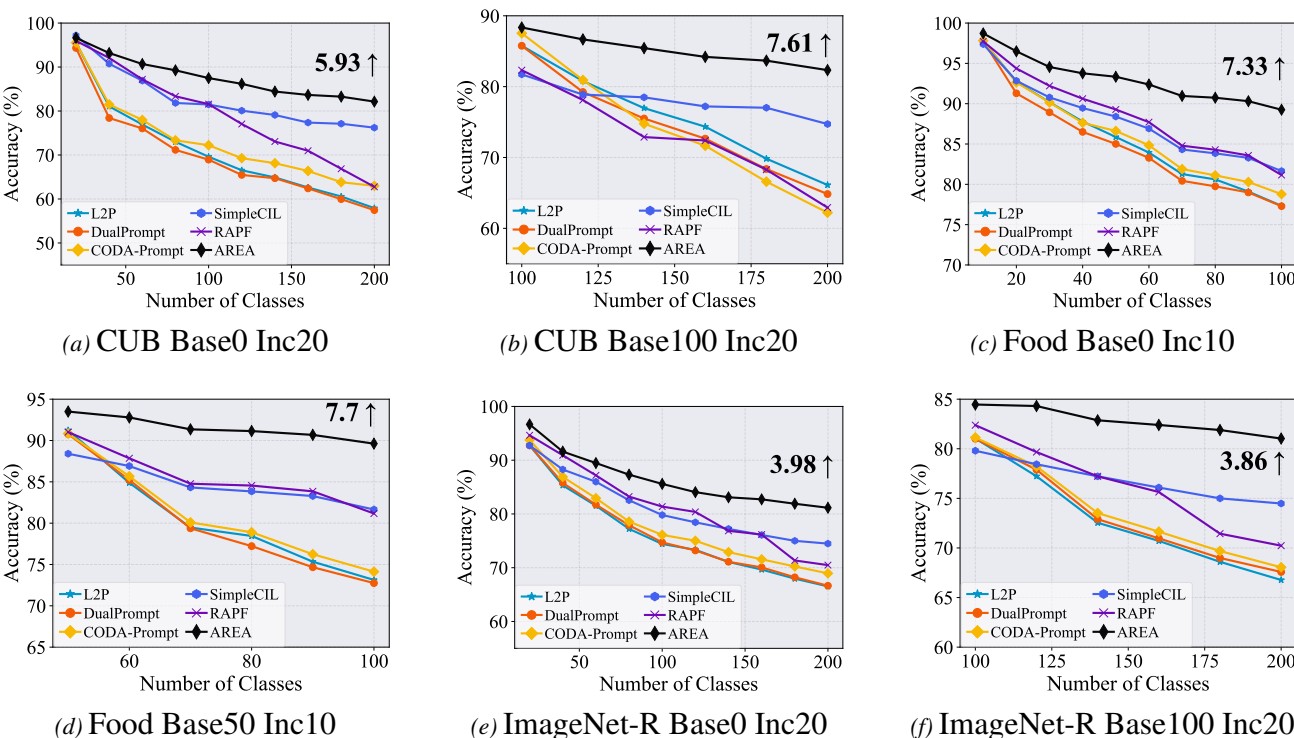

*(a)* CUB Base0 Inc20    *(b)* CUB Base100 Inc20    *(c)* Food Base0 Inc10

*(d)* Food Base50 Inc10    *(e)* ImageNet-R Base0 Inc20    *(f)* ImageNet-R Base100 Inc20

*Figure 9.* Incremental performance of different methods on CUB, Food-101, and ImageNet-R. We compare different base initialization settings (Base0 vs. Base100/50).

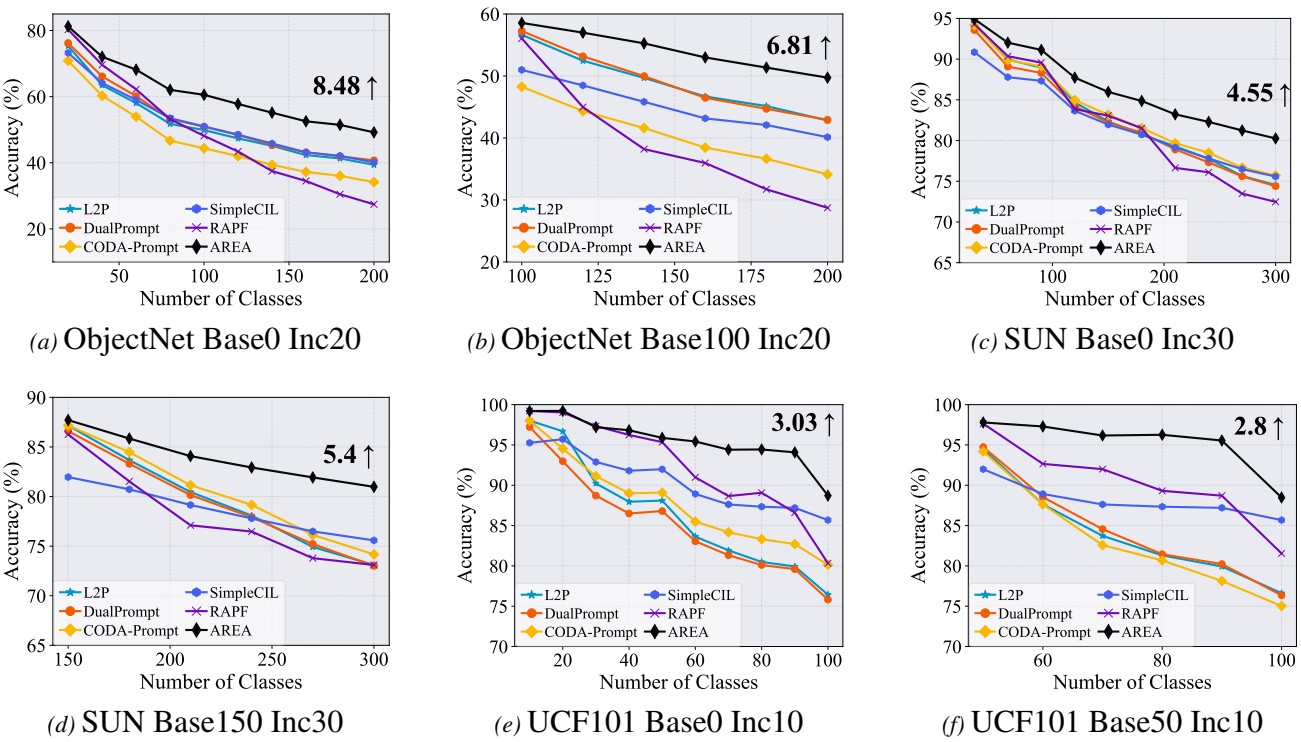

*Figure 10.* Incremental performance on ObjectNet, SUN397, and UCF101. The figures illustrate the effectiveness of AREA across various domain shifts and task lengths.

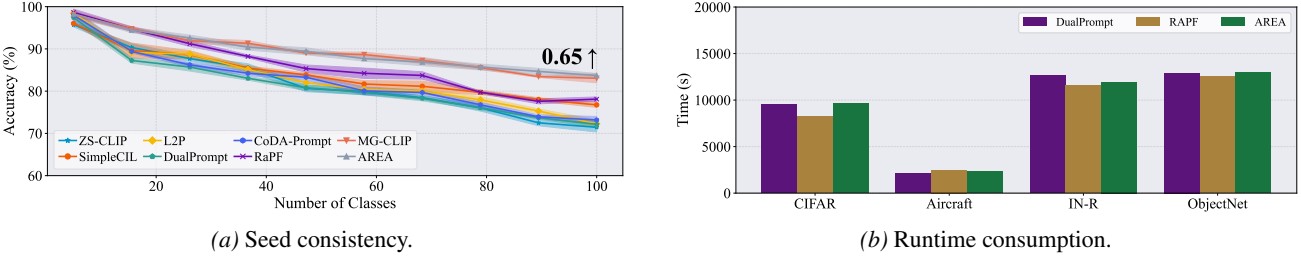

*(a)* Seed consistency.                    *(b)* Runtime consumption.

*Figure 11.* Additional analysis of AREA. Left: performance consistency across five random seeds on CIFAR-100 B0 Inc10, where the shaded area denotes the standard deviation. Right: runtime comparison across different settings.

slight overhead is primarily attributed to the additional computations required for PGA-based Attribute Extraction and the optimization process in OT-based Task Selection. However, considering the significant performance improvements achieved by these components, we argue that this modest increase in computational cost is well-justified. AREA maintains a favorable trade-off between efficiency and accuracy, remaining practically applicable for real-world incremental learning scenarios.

### E.4. Trainable Parameters

In this section, we further report the number of trainable parameters for each compared method in Tab. 5. As can be inferred from the results, AREA's trainable parameter quantity has the same scale as other compared methods, while AREA can achieve a better performance.

## F. Dependence on MLLM Annotation

We further analyze whether AREA strongly depends on powerful MLLMs and whether the noisy generated descriptions may affect performance. In addition to using GPT-5 captions, we evaluate a weaker open-source captioner, LLaVA-7B, and vary the proportion of captioned training samples. The comparison is reported in Tab. 6.

*Table 5.* Number of trainable parameters on CIFAR100 B0 Inc10 setting.

| Methods | Trainable Parameters |
|---|---|
| L2P (Wang et al., 2022b) | 0.16 M |
| DualPrompt (Wang et al., 2022a) | 0.33 M |
| CODA-Prompt (Smith et al., 2023) | 3.92 M |
| RAPF (Huang et al., 2024) | 0.26 M |
| MG-CLIP (Huang et al., 2025) | 0.58 M |
| **AREA** | 0.52 M |

*Table 6.* Effect of caption source and caption coverage on final accuracy.

| Caption source / coverage | CIFAR-100 | CUB-200 |
|---|---|---|
| Template only | 78.40 | 73.20 |
| LLaVA-7B, 20% samples captioned | 81.36 | 80.84 |
| LLaVA-7B, 100% samples captioned | 82.84 | 81.21 |
| GPT-5, 100% samples captioned | 83.69 | 82.14 |
| MG-CLIP | 82.78 | 64.25 |

As shown in Tab. 6, stronger MLLMs provide the best performance, but AREA is not overly dependent on a specific powerful captioner. Replacing GPT-5 with LLaVA-7B only leads to a small accuracy drop, and using LLaVA-7B captions for only 20% of the training samples still yields competitive results. Compared with template-only textual supervision, generated descriptions consistently improve performance, especially on fine-grained datasets such as CUB-200.

This robustness comes from the way AREA uses generated descriptions. Captions are not directly used as hard labels or final decision rules; instead, they are used to construct a textual attribute subspace. The final prediction is still based on coupled visual-textual aggregation and OT-based routing. Therefore, occasional caption noise can be absorbed by the attribute-level representation and has limited influence on the final classifier. These results indicate that AREA is both annotation-efficient and robust to moderate caption noise.

## G. Sensitivity to the Number of Attributes

We also study how the number of selected attributes affects performance. In practice, we use a shared moderate value of $K$ across datasets, selected by validation. Tab. 7 reports the sensitivity analysis on CIFAR-100 and ObjectNet.

As shown in Tab. 7, performance improves when $K$ increases from a small value to a moderate one, and then quickly saturates. On CIFAR-100, increasing $K$ from 8 to 16 brings only a marginal improvement, while on ObjectNet the performance slightly decreases when using $K = 16$. This suggests that too few attributes may be insufficient to describe class-level semantics, whereas too many attributes can introduce redundant or noisy directions. Based on this trend, we use $K = 8$ by default, which provides a good trade-off between expressivity and efficiency across datasets with different levels of granularity.

Adaptive selection of $K$ is a promising future direction. For example, one could allocate more attributes to visually diverse or fine-grained classes and fewer attributes to simpler classes. We leave this adaptive attribute allocation strategy for future work.

**Limitations.** Compared with replay-based class-incremental learning methods, AREA is more privacy-friendly because it is fully compatible with the exemplar-free setting and does not store old samples. Some strong prior methods, such as PROOF, require preserving exemplars from previous tasks, which may expose sensitive user data in real applications. In contrast, AREA avoids this issue by learning without retaining past samples. For more specialized settings such as long-tail, few-shot, or class-imbalanced continual learning, we do not include dedicated experiments in this work since they are beyond the main scope of this paper. Nevertheless, as AREA improves representation learning and decision robustness through occlusion-based intervention and multi-view augmentation, it has the potential to remain effective under these challenging scenarios. A more thorough study of these settings is left as future work.

*Table 7.* Sensitivity analysis of the number of attributes $K$.

| $K$ | CIFAR-100 | ObjectNet |
|---|---|---|
| 2 | 78.13 | 46.09 |
| 4 | 81.81 | 48.66 |
| 8 | 83.69 | 49.73 |
| 16 | 83.72 | 48.86 |

*Table 8.* Ablation study on PGA and OT-weighted mixture prediction.

| Variant | Aircraft attr.-text sim. | CIFAR-100 final acc. |
|---|---|---|
| PCA + full inference | 0.019 | 80.96 |
| PGA + sim only | 0.026 | 81.51 |
| PGA + single-task only | 0.026 | 82.24 |
| AREA full | 0.026 | 83.69 |

## H. Necessity of PGA and OT-weighted Mixture Prediction

We further ablate two key components of AREA: the geometry-aware PGA attribute construction and the OT-weighted mixture prediction in Eq. (18). These two components are complementary. PGA constructs stable class-level anchors on the hyperspherical CLIP embedding space, while the OT-weighted mixture prediction improves inference robustness when task manifolds partially overlap. Replacing PGA with Euclidean PCA ignores the intrinsic spherical geometry of normalized CLIP features. Similarly, replacing the soft mixture with a single-task or unweighted prediction rule makes inference more brittle near task boundaries.

Following this motivation, we evaluate several variants using two metrics: the semantic similarity between learned anchors and ground-truth attribute descriptions on Aircraft, and the final accuracy on CIFAR-100. The results are shown in Tab. 8.

As shown in Tab. 8, replacing PGA with PCA reduces both semantic alignment and final continual learning accuracy, demonstrating the importance of respecting the hyperspherical geometry of CLIP representations. The two simplified inference variants also underperform the full model. Using similarity-only prediction removes the routing confidence provided by OT, while single-task prediction discards useful collaboration among task-specific experts. In contrast, the full Eq. (18) leverages soft task weights to aggregate predictions across experts, which is especially beneficial when the test sample lies near overlapping task manifolds. These results confirm that both PGA and the OT-weighted mixture prediction are necessary for the final performance of AREA.

## I. Additional Discussion on Future Directions

Although AREA is currently instantiated for CLIP-based CIL, it may be extended in several directions. For tasks with larger modality gaps, future work could combine AREA with unified multimodal encoders such as Meta-Transformer (Zhang et al., 2023). For long-tailed, few-shot, or imbalanced continual learning, the masking and augmentation strategies may need to be adapted to avoid overfitting scarce classes. Finally, the ideas of attribute extraction, aggregation, and OT-based routing could be explored in larger MLLM continual learning frameworks, such as Mixture-of-LoRA-Experts.

## J. A Comprehensive Study on PTM-based Methods

This section summarizes the PTM-based Continual Learning approaches compared in our main paper. For a fair evaluation, all methods are implemented on top of the same frozen pre-trained backbone, and differ only in how they adapt or condition the model across tasks. The compared approaches are briefly described below:

- **L2P** (Wang et al., 2022b): This method proposes a prompt-pool mechanism for continual learning with a fixed backbone. For each incoming task, the model retrieves a small set of prompts via instance-wise querying, and uses these prompts to modulate the backbone while keeping its parameters unchanged. It operates without rehearsal and does not assume task identity at test time, making it effective in task-agnostic incremental scenarios.

- **DualPrompt** (Wang et al., 2022a): This method extends prompt pooling by disentangling prompts into shared

(task-invariant) prompts and task-specific prompts. The two prompt types provide complementary conditioning signals, enabling rehearsal-free adaptation while preserving prior knowledge, and yielding strong performance in class-incremental settings.

- **CODA-Prompt** (Smith et al., 2023): This method introduces a decomposed, attention-driven prompt construction strategy. Instead of selecting fixed prompts, it learns prompt components and composes them dynamically per input, improving adaptability to new classes while maintaining stability for previously learned classes.

- **SimpleCIL** (Zhou et al., 2025a): This method re-examines class-incremental learning with large pre-trained models, emphasizing generalization and adaptation as key factors. It provides simplified yet competitive baselines that serve as strong reference points in the PTM regime.

- **ZS-CLIP** (Radford et al., 2021): Although originally designed for zero-shot recognition, ZS-CLIP is included as a robust reference for incremental evaluation. It exploits CLIP's vision-language alignment to recognize novel classes without additional training.

- **RAPF** (Huang et al., 2024): This method presents a CLIP-based class-adaptive prompt fusion scheme, where prompts for new classes are adapted and fused with existing prompts to accommodate sequential class acquisition.

- **MG-CLIP** (Huang et al., 2025): This method develops a multi-granularity incremental framework for CLIP that jointly models class-level and instance-level changes, and mitigates modality mismatch when classes are introduced over time.

