# OpenReview forum: "AREA: Attribute Extraction and Aggregation for CLIP-Based Class-Incremental Learning"
_ICML.cc/2026/Conference — ICML 2026 regular_

### Official Review · Reviewer_xnSd · 2026-02-20

**Soundness:** 3
**Presentation:** 3
**Significance:** 3
**Originality:** 3
**Overall Recommendation:** 4
**Confidence:** 5

**Summary:**

In this paper, authors proposed CLIP-based CIL from the perspective of attribute extraction and aggregation, where class-level visual and textual attributes are obtained in the hyperspherical embedding space, and then these attributes are aggregated based on learnable task-specific score mapping and residual refinement mapping. During inference, weighted mixture of all tasks’  image-text alignments based on optimal transport is adopted for class prediction. Experiments on 11 benchmarks confirm the efficacy of the proposed method.

**Compliance With Llm Reviewing Policy:**

Affirmed.

**Final Justification:**

Authors have well addressed most of my concerns and promise to revise the manuscript for the remaining concerns. Therefore, I decide to keep the original recommendation "Weak Accept".

**Key Questions For Authors:**

1.	Please perform further ablation studies as mentioned in weakness 1&2 to check whether the proposed computing components are really necessary.
2.	Please provide evidence for the attribute interpretation of the computed anchors.
3.	Please clarify how to set these values for each benchmark.

**Limitations:**

Yes

**Strengths And Weaknesses:**

Strengths:

1.	The overall idea and framework are novel, with each component well justified in methodology.

2.	Extensive empirical evaluations were performed, consistently supporting the superior performance of the proposed method compared to multiple strong baselines.

3.	The paper is well presented and easy to follow.

Weaknesses:

1.	The necessity of the using principal geodesic analysis (PGA) for attribute construction is not empirically supported. Fine-grained ablation study of this part should be performed, e.g., reporting CIL performance after replacing PGA-based method by PCA-based method to construct attributes.

2.	The mixture of experts for model prediction (Eq. 18) is not well justified. First, if class c belongs to task b, why is the term p*sim from other tasks helpful to help choose the class c? Second, fine-grained ablation study should be performed, e.g., (1) removing the summation and only using the task b which contains the class c, and (2) only using the sim without the p(b|x) term.

3.	While “attributes” are used throughout the paper, there is no any theoretical and empirical evidence to support that the obtained anchors really correspond to semantic attributes.

4.	Some hyperparameter settings are not mentioned, particularly for lambda_int and lambda_comp. From Section 6.3, it seems that values these hyperparameters are chosen based on grid-search on each test set, which should be forbidden.

5.	Dimensionality of some variables are questionable. For example, dimension of S_b in Eqs (7)(8) probably should be K-by-d rather than d-by-K; Also based on definition of S_b, the difference term (between two s) in Eq. (12) is a vector rather than scalar, therefore the definition of Eq. (12) as a scalar loss is probably erroneous.

6.	The proposed method depends on pretrained and fixed image and text encoder which are powerful enough to extract discriminative attributes for all tasks. This may not be true if data modality of downstream continual learning tasks is clearly different from that of the dataset for encoder pretraining.

---

> ### Author Rebuttal · Authors · 2026-03-31
>
> ### W1, W2 & Q1: necessity of PGA and the mixture prediction in Eq. (18)
> The necessity of both geometry-aware attribute construction and soft mixture prediction lies in their coupling: PGA builds stable class-level anchors on the hyperspherical CLIP space, while the OT-weighted mixture in Eq. (18) improves robustness under partial task-manifold overlap. Replacing PGA with PCA ignores spherical geometry; replacing the soft mixture with a single-task or unweighted rule makes prediction more brittle. In short, PGA stabilizes extraction, and the OT-weighted mixture stabilizes inference.
>
> Following the reviewer's suggestion, we performed the requested ablations. To evaluate these components, we report two metrics: the semantic similarity between the learned anchors and ground-truth attribute descriptions on the Aircraft dataset, and the final accuracy on CIFAR-100.
>
> |Variant| Aircraft attr.-text sim.|CIFAR-100 final acc.|
> |-|-|-|
> | PCA + full inference|0.019 |80.96 |
> | PGA + sim only (remove $p(b\mid x)$)|0.026 |81.51 |
> | PGA + single-task only (remove summation)|0.026 |82.24 |
> | AREA (full)|**0.026** |**83.69** |
>
> Replacing PGA with PCA/SVD consistently reduces semantic alignment and CIL accuracy, showing the benefit of hyperspherical geometry. Both simplified inference variants also underperform full Eq. (18): removing $p(b|x)$ loses routing confidence, while using only the nominal-task expert loses useful soft collaboration near task boundaries. Thus, the full model is empirically necessary.
>
> ### W3 & Q2: evidence for the attribute interpretation of the anchors
>
> We do not claim that each anchor is a manually labeled symbolic attribute; rather, the anchor subspace captures semantically meaningful fine-grained directions. **Fig. 4a in the main paper** already shows that PGA-based anchors align better with fine-grained textual descriptions than raw image features or SVD-based anchors. As an additional probe, we search in the CLIP-vocabulary codebook to find the nearest token for the learned anchors of each class and show the results partially below.
>
> | #| PGA anchors (Similarity)   | SVD anchors (Similarity)  |
> |-|-|-|
> | 1 (envelope)| `red (0.2436)`, `inscription (0.2229)`      | `flour (0.2144)`, `phx (0.2030)`  |
> | 2 (hand mirror)| `pavement (0.2085)`, `tan (0.1844)`         | `ube (0.1986)`, `gbbo (0.1983)`   |
> | 3 (jeans)| `blueberries (0.2805)`, `bluejays (0.2332)` | `ube (0.2562)`, `copper (0.2478)` |
>
> Although indirect, this probe is still informative: PGA anchors are much closer to fine-grained, category-relevant concepts, whereas SVD anchors are often arbitrary subwords or weakly related tokens. Together with **Fig. 4a**, this supports the semantic meaningfulness of the learned anchor directions.
>
> ### W4 & Q3: how $\lambda_{int}$ and $\lambda_{comp}$ are set
>
> **The values of $\lambda_{int}$ and $\lambda_{comp}$ were not selected by grid-search on each test set. Fig. 3b in the main paper is only a sensitivity analysis. In practice, we choose $(\lambda_{int}, \lambda_{comp})=(0.8,1.0)$ once and keep it fixed across benchmarks.** We also tested a conservative setting using the worst pair from the sweep.
>
> |Setting | CIFAR-100 |   Food101 |
> |-|-|-|
> |RAPF|77.93|81.15|
> |AREA, fixed $(0.8,1.0)$|**83.69**|**89.25**|
> |AREA, worst pair $(0.5,0.5)$|82.88|85.04|
>
> Even with the worst pair, AREA remains above the strongest prior baseline.
>
> ### W5: dimensionality / notation issues in Eqs. (7), (8), and (12)
>
> Thanks for your suggestion, this is a notation issue. The score mapping $S_b$ produces a $K$-dimensional score vector (equivalently, use a $K \times d$ matrix), which is then multiplied with the anchored basis. Likewise, Eq. (12) should be understood as an elementwise positive-part operation followed by scalar reduction, e.g., $\|\max(0, s(\tilde{x}_i)-s(x_i))\|_1$. We will correct these expressions in the camera-ready version.
>
> ### W6: dependence on fixed pretrained encoders
>
> We agree with the reviewer that this assumption may not hold when downstream continual learning tasks involve modalities clearly different from those seen in encoder pretraining. In this work, both our method and the compared baselines focus on a well-defined **CLIP-based class-incremental learning** setting, i.e., using CLIP for continual new-class learning in the **image modality**. For tasks with larger modality gaps or other modalities, one may first build stronger pretrained representations using modality-specific encoders or unified multimodal embeddings, e.g., Meta-Transformer [1], and then apply AREA on top for continual learning. We will clarify this scope and discuss this extension as future work in the camera-ready version.
>
> [1] Meta-Transformer: A Unified Framework for Multimodal Learning. arXiv:2307.10802, 2023.

---

> > ### Author Rebuttal · Reviewer_xnSd · 2026-04-03
> >
> > Most of my concerns have been resolved! However, the response to the attribute issue is still not convincing enough. Fig 4a is based on relative comparison, and the results in rebuttal imply that the "attributes" are actually not well aligned with real visual feature descriptions. Therefore, I strongly suggest that the authors include a discussion in the final version to clarify that the attributes mentioned in this work may not exactly correspond to certain exact attributes of classes. Please also include the provided empirical justifications to the final version.

---

> > > ### Author Response · Authors · 2026-04-03
> > >
> > > We thank the reviewer for acknowledging that most of their concerns have been addressed.
> > >
> > > We will explicitly clarify in the camera-ready version that the attributes we refer to in our work may not correspond exactly to the attributes of the classes. Instead, the learned anchors represent generalizable fine-grained directions in the embedding space, which are more abstract than traditional, manually labeled attributes. We will also incorporate the empirical justification provided, including the results from Fig. 4a in the main paper and the nearest-token probe we conducted. This distinction will be made clearer to avoid any confusion.
> > >
> > > Thank you again for taking the time to review this paper.

---

### Official Review · Reviewer_eeuA · 2026-03-12

**Soundness:** 3
**Presentation:** 2
**Significance:** 2
**Originality:** 2
**Overall Recommendation:** 4
**Confidence:** 4

**Summary:**

The paper proposes AREA, a method designed to improve the accuracy and stability of class-incremental learning. It first employs principal geodesic analysis to enhance the stability of attribute extraction. For attribute aggregation, the paper introduces lightweight task-specific experts to learn more generalizable and robust representations, mitigating the problem of “aggregation drift,” and further simplifies prediction through routing over task attribute manifolds. During inference, Optimal Transport is used to compute the distributional distance between a query input and the attribute manifold of each task. Experiments on multiple datasets demonstrate that AREA effectively alleviates catastrophic forgetting and other challenges in CLIP-based class-incremental learning.

**Compliance With Llm Reviewing Policy:**

Affirmed.

**Key Questions For Authors:**

1.What is the necessity of attribute extraction and attribute aggregation for optimizing CLIP-based CIL? Could the variation in attribute aggregation within the same class introduce noise? Given that there are existing studies on attribute extraction and aggregation in CLIP-based CIL, does AREA introduce any methodological novelty?
2.Although Table 2 demonstrates the method’s robustness across different MLLMs, this introduces a reliance on powerful external models. Could this limit the applicability of AREA in scenarios where strong MLLMs are not available? Additionally, does relying on MLLM-generated descriptions potentially introduce noise?
3.For highly variable tasks, is the proposed occlusion strategy still applicable? A clarification or discussion would be helpful.
4.In the paper, the number of attributes K is an important hyperparameter. How is K selected? Is it sensitive to different types of datasets? Is there a method to adaptively determine the optimal value of K?

**Limitations:**

The paper does not provide a detailed discussion of the limitations of AREA. The authors could consider supplementing this by exploring the following aspects:
1.Regarding the privacy issues mentioned in SOTA methods in the Related Work, does AREA offer any improvements?
2.For class-imbalanced datasets, such as those with long-tail distributions or few-shot categories, is AREA applicable and effective?

**Strengths And Weaknesses:**

a)Soundness: The paper is technically well-founded. It uses PGA to stabilize attribute extraction, employs VIB to ensure aggregation generalization, and applies OT for expert routing during inference. Each design choice is well-matched to the problem it addresses and theoretically justified. The experiments consider robustness with respect to both hyperparameters and dataset variations. However, the authors do not provide an evaluation of AREA’s potential limitations.
b)Presentation: The paper is clearly structured, but it lacks a detailed discussion of the shortcomings in current CLIP-based CIL research and how AREA specifically improves upon existing methods. Missing the introduction of the article structure.
c)Significance: Addressing catastrophic forgetting in CLIP-based CIL allows the model to retain previously learned knowledge while learning new tasks, improving stability and generalization, and fully leveraging CLIP’s multi-modal representations. This enables high performance and practical applicability in long-term incremental learning with large-scale data and many unseen categories. However, research on attributes in current SOTA CLIP-based CIL methods is already relatively mature, and AREA’s methodological improvements are somewhat limited, which constrains its overall contribution to the field.
d)Originality: AREA enhances the interpretability of CLIP embeddings through attribute extraction and aggregation. The use of optimal transport theory to improve inference accuracy is also innovative. While the paper introduces theoretical and conceptual contributions, it does not propose new datasets or tasks, and it lacks a clear evaluation of existing method limitations or an explicit comparison of its own novelty.

---

> ### Author Rebuttal · Authors · 2026-03-31
>
> ### Q1. Necessity; Noise; Novelty
> We thank the reviewer for this important question.
>
> Regarding necessity and novelty, our key motivation is that forgetting in CLIP-based CIL does not arise from a single similarity shift, but from two coupled drifts: extraction drift, where the model changes the cues used to represent old classes, and aggregation drift, where it changes how these cues are combined as new tasks arrive. AREA is designed around this view, with PGA-based hyperspherical anchors to stabilize extraction, and task-specific scorers, VIB regularization, and OT-based routing to stabilize aggregation and inference. Therefore, our novelty is not simply introducing attributes, but providing a unified drift-oriented framework that connects motivation, training, and inference under the perspective of attribute-evidence drift.
>
> Regarding intra-class noise, unconstrained aggregation can indeed become noisy if it overfits task-specific shortcuts. AREA addresses this with VIB-style regularization and intervention-based monotonicity, which suppress unstable cues and preserve view-invariant, label-relevant evidence. Empirically, variance across random seeds is small, and **Fig. 10 in the main paper** also shows stable rather than noisy score patterns.
>
> |CIFAR-100 B0 Inc10|Mean final acc.|Std. over 5 seeds|
> |-|-|-|
> |w/o VIB regularization|82.71|0.84|
> |AREA|83.69|0.29|
>
> These results suggest that attribute aggregation in AREA does not inject harmful noise; instead, the regularized aggregation makes prediction more stable.
>
> ### Q2. Dependence on strong MLLMs and possible noise in generated descriptions
>
> We agree this is an important concern. Our results suggest that AREA is not strongly dependent on a specific powerful MLLM: replacing GPT-5 with LLaVA-7b causes only a small drop. AREA is also annotation-efficient, retaining good performance even when only a subset of samples is captioned.
>
> |Caption source / coverage|CIFAR-100 final acc.| CUB-200 final acc.
> |-|-|-|
> |Template only (no extra descriptions) |78.40|73.20|
> |LLaVA, 20% samples captioned|81.36|80.84|
> |LLaVA, 100% samples captioned|82.84|81.21|
> |GPT-5, 100% samples captioned|**83.69**|**82.14**|
> |MG-CLIP|82.78|64.25|
>
> As shown in the above table, while descriptions from strong MLLMs yield optimal performance, AREA degrades gracefully with smaller models (LLaVA-7b) or partial caption coverage. Possible caption noise is also mitigated because captions are only used to build a textual attribute subspace, and final prediction still relies on coupled visual-textual aggregation and OT routing.
>
> ### Q3. Is the occlusion strategy still applicable for highly variable tasks?
>
> Yes. The benchmarks we tested on already include highly variable datasets such as ObjectNet. These datasets exhibit not only significant intra-class variations but also large differences in the categories learned across different tasks. Our occlusion strategy is not meant to model all task variability, but to prevent the scorer from spuriously increasing confidence under corrupted inputs and thus discourage shortcut evidence, thereby achieving competitive results even on these highly variable datasets.
>
> ### Q4. How is the number of attributes $K$ selected? Is it sensitive? Can it be adaptive?
>
> In practice, we choose a shared moderate value of $K$ by validation and use it across datasets. Performance improves from small $K$ to moderate $K$, then quickly saturates, indicating low sensitivity.
>
> |$K$|CIFAR-100| ObjectNet|
> |-|-|-|
> |2|78.13|46.09|
> |4|81.81|48.66|
> |8|83.69|49.73|
> |16|83.72|48.86|
>
> Based on this trend, we use **$K=8$** by default, as it offers a good trade-off between expressivity and efficiency across datasets of different granularity. Adaptive $K$ selection is an interesting future direction, e.g., allocating $K$ by class/task complexity, and we will consider this as a direction for future work.
>
> ### Limitations
>
> Compared with replay-based CIL methods, AREA is more privacy-friendly because it is fully compatible with the exemplar-free setting and does not store old samples. In contrast, some strong prior methods, such as PROOF [1], require preserving exemplars from previous tasks, which may expose sensitive user data and thus raise privacy concerns in real applications. AREA avoids this issue by learning without retaining past samples. As for more challenging settings such as long-tail, few-shot, or class-imbalanced continual learning, we do not include dedicated experiments in this paper because these are specialized problem settings beyond the current scope. Nevertheless, since AREA improves representation learning and decision robustness through occlusion and multi-view augmentation, it has the potential to remain effective in these scenarios. We will clarify this point and leave a more thorough study under these settings as future work.
>
> [1] Learning without forgetting for vision-language models. TPAMI, 2025.

---

> > ### Author Rebuttal · Reviewer_eeuA · 2026-04-04
> >
> > The authors have provided satisfactory responses to all my questions. The VIB ablation and seed-variance results address my concern about intra-class noise in attribute aggregation (Q1). The MLLM replacement experiment (GPT-5 → LLaVA-7b, 100% → 20% coverage) convincingly shows that AREA is not critically dependent on strong external models (Q2). The clarification on the occlusion strategy (Q3) and the K sensitivity ablation (Q4) are also reasonable. I will maintain my original score.

---

> > > ### Author Response · Authors · 2026-04-04
> > >
> > > Thank you for your kind response. We are glad to hear that our clarifications and experiments addressed your concerns. We sincerely appreciate the time and effort you devoted to reviewing our work.

---

### Official Review · Reviewer_CuVC · 2026-03-12

**Soundness:** 3
**Presentation:** 3
**Significance:** 2
**Originality:** 2
**Overall Recommendation:** 4
**Confidence:** 4

**Summary:**

This paper proposes AREA, a CLIP-based class-incremental learning method that reduces catastrophic forgetting by decomposing classification into attribute extraction and attribute aggregation. Its main contribution is to stabilize both stages with attribute anchors, task-specific aggregation, and test-time expert routing, giving a more structured way to preserve old knowledge while learning new classes.

Experimentally, the method is evaluated on nine benchmarks and consistently outperforms prior CLIP-based continual learning baselines. The gains are often substantial, especially on challenging settings such as ObjectNet and Aircraft, while also remaining strong on datasets like SUN and Cars. The ablation studies further show that each component contributes to performance, and the method appears robust even when annotation coverage is reduced. Overall, the paper’s main strength is turning CLIP-based continual learning into an attribute-centric framework and backing it up with strong empirical results.

**Compliance With Llm Reviewing Policy:**

Affirmed.

**Final Justification:**

My original concerns have been fully resolved. I will keep my score at Weak Accept.

**Key Questions For Authors:**

#1. Inference efficiency: AREA adds OT-based routing over all past tasks and then aggregates predictions from task-specific experts at test time. Could the authors report explicit inference latency, memory overhead, and scaling with the number of tasks/classes, since the paper mainly emphasizes accuracy gains and only briefly mentions runtime?

#2. Why does AREA fail less than MG-CLIP on some splits? In Table 1, MG-CLIP appears to collapse badly on several settings while AREA stays much more stable. Do the authors have a concrete explanation for what makes AREA especially robust in those harder incremental regimes?

#3. Contribution of OT routing vs. simpler routing rules: The ablation shows that adding OT improves performance, but could the authors compare against simpler inference schemes such as nearest-task or cosine-based routing with matched compute? That would make it easier to judge whether the gain comes from distributional routing itself or from added inference complexity.

**Limitations:**

My main reservation is that the problem setting considered in the paper is somewhat limited in scope. As discussed in the Weaknesses section, multimodal-LLM-based class-incremental learning is being actively explored and may be more aligned with the current frontier of the field. However, I do not view this as a fatal issue, since more classical formulations can still provide meaningful technical insights and practical value. For this reason, although the scope is somewhat limited, I lean toward a Weak Accept.

**Strengths And Weaknesses:**

**Strengths**

A key strength of the paper is its coherent formulation of the proposed framework. The authors explicitly disentangle extraction drift and aggregation drift, and each of Secs. 4.1–4.3 is designed to address one part of this problem in a principled manner. While some components are still approximations, the overall pipeline is theoretically well motivated and translated into an optimization procedure that appears practical and effective.

Another strength of the paper is the overall quality of the empirical evaluation. The method is tested on nine benchmarks and shows consistently strong gains over prior CLIP-based continual learning baselines, with improvements that are not limited to a single dataset or split. The reported learning curves are also convincing in that they suggest smoother degradation over time, which supports the paper’s claim that the method mitigates catastrophic forgetting rather than merely improving a few endpoint numbers.

**Weaknesses**

A main limitation is the scope of the paper: although the method incorporates MLLM-generated captions, the proposed learner itself is still specialized to frozen CLIP-based class-incremental classification rather than continual learning of multimodal LLMs. The paper also explicitly acknowledges that AREA has only been instantiated for CLIP-based CIL. As a result, the strong empirical results mainly support effectiveness in the CLIP-style classification setting, while it remains unclear whether the proposed extraction/aggregation decomposition and routing strategy would transfer to modern instruction-tuned or generative MLLMs, where the objectives and forgetting behaviors can be substantially different.

---

> ### Author Rebuttal · Authors · 2026-03-31
>
> ### Q1: Explicit inference latency, memory overhead, and scaling with the number of tasks/classes
> We agree that explicit efficiency reporting is necessary. We summarize end-to-end average inference cost and peak VRAM overhead under three class scales, using CIFAR-100, CUB-200, and SUN-300 (the first 300 classes of SUN-397), all on the same RTX 4090 with batch size 1.
>
> | Setting   | #Classes | AREA (ms/sample) | AREA w/o OT (ms/sample) | Cosine Routing (ms/sample) |
> | -| --| -| -| -|
> | CIFAR-100 | 100      | 16.4             | 10.7                    | 13.5                       |
> | CUB-200   | 200      | 17.7             | 10.8                    | 14.4                       |
> | SUN-300   | 300      | 18.2             | 12.2                    | 15.6                       |
>
> | Memory Overhead\ # Classes | 100     | 200     | 300     |
> | -| -| -| -|
> | AREA (OT Routing)          | ~1.6 MB | ~3.2 MB | ~4.7MB  |
> | Cosine Routing             | ~0.16MB | ~0.32MB | ~0.49MB |
>
> These results show two key points. First, AREA scales mildly from **16.4 ms** to **18.2 ms** when moving from **100** to **300** classes, suggesting that routing remains nearly flat in practice. This is because AREA routes over compact task-level attribute manifolds with batched Sinkhorn iterations, rather than exhaustive class-wise matching. Second, Although the memory overhead required by AREA grows almost linearly, in practice, because the constant factor is very small, it only requires less than 5MB of additional overhead even for 300 classes, which is completely acceptable. Thus, OT routing adds a small constant overhead but does not become a practical bottleneck as tasks/classes grow.
>
> ### Q2: Why is AREA more robust than MG-CLIP on several challenging splits?
>
> Our main point is that AREA is an integrated design for the two failure modes identified in the paper: attribute extraction drift and attribute aggregation drift. PGA-based anchors stabilize what attributes are extracted; the scorer/residual module together with VIB regularization stabilizes how these attributes are aggregated; OT routing then uses these anchored task manifolds to reduce cross-task interference at inference. These components are tightly coupled rather than independent add-ons.
>
> By contrast, MG-CLIP is more fragile on hard splits mainly because it keeps the shared representation space too conservative. In difficult or fine-grained later tasks, this limits plasticity. When similar new classes arrive, the rigid old decision structure makes feature collisions more likely. AREA avoids this failure mode by combining stable class-level anchors with lightweight task-local experts: the former preserve old knowledge, while the latter still allow enough flexibility to fit subtle new decision boundaries. This is also consistent with the strong gains of AREA on difficult benchmarks reported in the paper.
>
> ### Q3: Why is OT-based routing preferable to simple cosine-similarity matching?
>
> We agree that cosine matching is a natural baseline. We therefore compare OT routing and cosine routing directly on CIFAR-100 at different incremental stages of CIFAR100 b0-Inc10.
>
> | Incremental Stage                           | Task 2 | Task 5 | Task 10 |
> | -| - | - | - |
> | AREA w/ OT: latency (ms/sample)             | 16.5   | 16.2   | 16.4    |
> | AREA w/ cosine routing: latency (ms/sample) | 13.7   | 13.2   | 13.5    |
> | AREA w/ OT: accuracy (%)                    | 94.00  | 89.08  | 83.69   |
> | AREA w/ cosine routing: accuracy (%)        | 93.80  | 87.04  | 80.30   |
>
> The trade-off is favorable. At the final 100-class stage, OT routing adds only **3 ms/sample** over cosine routing, but improves final accuracy by **+3.39%**. The reason is that cosine routing is a point-to-point comparison and is therefore sensitive to local feature drift, while our OT routing compares the query against the whole task attribute manifold. This distribution-aware matching is more robust when different tasks partially overlap in the CLIP embedding space. Hence, OT is not introduced for complexity’s sake; it is precisely what reduces cross-task misrouting in the difficult long-horizon regime.
>
> ### Weaknesses: Applicability to MLLMs
>
> Admittedly, applying the entire AREA framework directly to MLLM continual learning is non-trivial. However, we argue that its core modules, with slight adaptations, offer highly transferable insights, particularly for the mainstream Mixture-of-LoRA-Experts architectures currently used in MLLM continual learning. Specifically, our feature extraction and aggregation mechanisms can assist the MoLE Router in better capturing task-specific features, thereby identifying task similarities to facilitate cross-task knowledge sharing and redundant parameter pruning. Furthermore, our proposed OT-based routing strategy can effectively mitigate the common issue of routing failure. We will therefore present these specific adaptations as a promising future direction in the camera-ready version.

---

> > ### Author Rebuttal · Reviewer_CuVC · 2026-04-03
> >
> > My original concerns have been fully resolved. I will keep my score at Weak Accept.

---

> > > ### Author Response · Authors · 2026-04-03
> > >
> > > Thank you for your kind response. We are glad to hear that our clarifications addressed your concerns. We sincerely appreciate the time and effort you devoted to reviewing our work.

---

### Decision · Program_Chairs · 2026-04-30

**Decision:**

Accept (regular)

**Comment:**

This paper proposes AREA, a method for class-incremental learning based on CLIP. The core idea is to decompose classification into two stages: attribute extraction and attribute aggregation, to mitigate catastrophic forgetting. The method stabilizes attribute extraction by anchoring class-level visual and textual attributes in the hyperspherical embedding space via PGA, and stabilizes attribute aggregation by learning lightweight task-specific experts regularized by a VIB objective. During inference, it performs routing over task attribute manifolds using OT. Experiments on nine benchmarks show consistent improvements over prior CLIP-based continual learning methods.

The initial concerns from the reviewers focused on inference efficiency, practical robustness, the necessity of core components such as PGA and OT routing, dependence on external MLLMs, and the semantic grounding of the extracted attributes. After the rebuttal, two reviewers’ concerns have been fully resolved, and one reviewer’s concern has been partially resolved.

Overall, this work meets the requirements of the conference, but the authors should revise the manuscript to address the remaining issue, especially the clarification of attribute interpretation. For these reasons, I recommend acceptance.